
# STUDY OF AFRICAN DUST WITH MULTI-WAVELENGTH RAMAN LIDAR DURING THE "SHADOW" CAMPAIGN IN SENEGAL

I. Veselovskii[1,2], P. Goloub[3], T. Podvin[3], V. Bovchaliuk[3], Y. Derimian[3], P. Augustin[4], M. Fourmentin[4], D.Tanre[3], M. Korenskiy[1], D.N. Whiteman[5], A. Diallo[6], T. Ndiaye[6], A. Kolgotin[1], O. Dubovik[3]

[1]*Physics Instrumentation Center of GPI, Troitsk, Moscow, 142190, Russia*

[2]*Joint Center for Earth Systems Technology, UMBC, Baltimore, MD, USA*

[3]*Laboratoire d'Optique Atmosphérie, Université de Lille-CNRS, 59650, Villeneuve d'Ascq, France*

[4]*Laboratoire de Physico-chimie de l'atmosphère, Université du littoral côte d'Opale, France*

[5]*NASA GSFC, Greenbelt, MD 20771, USA*

[6]*Institut de Recherche pour le Développement, Dakar, Sénégal*

**ABSTRACT**
West Africa and the adjacent oceanic regions are very important locations for studying
dust properties and their influence on weather and climate. The SHADOW (Study of SaHAran
Dust Over West Africa) campaign is performing a multi-scale and multi-laboratory study of
aerosol properties and dynamics using a set of in situ and remote sensing instruments at an
observation site located at IRD (Institute for Research and Development) Center, Mbour,
Senegal ($14^0$N, $17^0$W).   In this paper, we present the results of lidar measurements performed
during the first phase of SHADOW which occurred in March-April, 2015. The multiwavelength
Mie-Raman lidar acquired $3\beta+2\alpha+1\delta$ measurements during this period. This set of measurements
has permitted particle intensive properties such as extinction and backscattering Ångström
exponents (BAE) for 355/532 nm wavelengths corresponding lidar ratios and depolarization ratio
at 532 nm to be determined. The mean values of dust lidar ratios during the observation period
were about 53 sr at both 532 nm and 355 nm, which agrees with the values observed during the
SAMUM 1 and SAMUM 2 campaigns held in Morocco and Cape Verde in 2006, 2008. The
mean value of particle depolarization ratio at 532 nm was 30±4.5%, however during strong dust
episodes this ratio increased to 35±5%, which is also in agreement with the results of the
SAMUM campaigns. The backscattering Ångström exponent during the dust episodes decreased





to ~-0.7, while the extinction Ångström exponent though being negative, was  greater than -0.2.
Low values of BAE can likely be explained by an increase in the imaginary part of the dust
refractive index at 355 nm compared to 532 nm. The dust extinction and backscattering
coefficients at multiple wavelengths were inverted to the particle microphysics using the
regularization algorithm and the model of randomly oriented spheroids. The analysis performed
has demonstrated that the spectral dependence of the imaginary part of the dust refractive index
may significantly influence the inversion results and should be taken into account.
**1. INTRODUCTION**

10        The impact of desert dust emitted into atmosphere on the Earth's radiation budget is the

subject of intense research (Sokolik and Toon, 1996;  Balkanski et al., 2007; Mahowald et al.,
2010; Formenti et al., 2011, 2014). Due to the wind patterns involved,  dust can be transported
far away from the main source regions in Africa and Asia allowing dust to be distributed in
varying amounts  all over the globe. North Africa is the largest source of dust in the world and
several field campaigns have been conducted to evaluate  dust particle microphysical properties
over Western Africa and to study long range transport of Saharan dust  (Reid et al., 2003; Tanre
et al., 2003; Redelsperger et al., 2006; Haywood et al., 2008; McConnell et al., 2008). During
these campaigns, dust particles were studied via aircraft, ground sampling and using sun
photometer measurements. However, vertical distribution of dust has received little attention
even though dust vertical structure is critical for an improved understanding of dust advection,
transport and dust-cloud interactions. The commonly used instrument to evaluate the height
profile of dust particle properties is the aerosol lidar. The numerous measurements performed in
Europe, America and Asia with multiwavelength Raman and HSRL lidar systems have resulted
in a significant amount of information about the vertical distribution of  dust intensive properties,
such as depolarization, lidar ratios, extinction and backscattering Ångström exponents (Sakai et
al., 2003; De Tomasi et al., 2003; Shimizu et al., 2004; Mona et al., 2006; Papayannis et al.,
2008; Xie et al., 2008; Ansmann et al., 2012; Burton et al., 2014; Nisantzi et al., 2015). However
these measurements were mostly performed at a significant distance from the source area, so the
dust particles were aged due to mixing with local aerosols and coating with soluble aerosol
species (Li et al., 2009) and may not have well represented the characteristics of the dust upon
initial emission.





To analyze the properties of pure dust measurements near the source regions are needed.
Such measurements of Saharan dust were performed during the SAMUM1 and SAMUM2
experiments using the assembly of Raman and HSRL lidars (Ansmann et al., 2011). During
those measurements the dust episodes and more complicated events, when the dust and smoke
layers occurred simultaneously, were studied (Tesche et al., 2009a,b; 2011; Esselborn et al.,
2009). However, for the estimation of aerosol radiative forcing not only the particle intensive
parameters, but also their microphysical properties, such as size, concentration and the complex
refractive index (CRI) are needed. An estimation of the vertical distribution of particle
microphysics can be achieved, for example, by combining lidar and sun photometer
measurements; a review of such studies can be found in a recent publication (Binietoglou et al.,
2015). However in these retrievals the mean radii and refractive indices of particles in the fine
and the coarse mode are assumed to be height independent, and only particle volume in each of
the modes is permitted to vary. Such assumptions may become invalid when aerosol layers of
different origins occur.
The alternative approach to evaluating the vertical distribution of dust properties is to
estimate the particle properties from lidar measurements only. Raman (or HSRL)
multiwavelength lidars based on a tripled Nd:YAG laser are able to provide three particle
backscattering and two extinction coefficients (so called $3\beta+2\alpha$ dataset). Different techniques
have been considered to invert these measurements into particle microphysics (Ansmann and
Müller, 2005), but the main issue is small number of input measurements (typically five),
compared to the numerous parameters needed for describing the aerosol microphysical
properties. This implies that the inverse problem is underdetermined and that numerous solutions
may reproduce the input measurements with similar accuracy. This family of solutions can be
localized by applying constraints to the "search space", i.e. limiting the range of particle radii and
refractive indices considered. The additional assumption usually made is that the refractive index
is spectrally independent and identical over the whole size range (Müller et al., 1999;
Veselovskii et al., 2002). Such an approach has proved to be efficient for aerosol particle size
distributions (PSD) with a predominant fine mode as, for example, in the case of biomass
burning aerosols (Müller et al., 2005; Veselovskii et al., 2015). However, in the case of dust the
inversion of lidar measurements becomes more challenging since the dust PSD contains a strong
coarse mode with particle radii extending up to ~ 15 µm and the estimation of properties for such



big particles is less accurate when measurements are only performed in the wavelength range of
355-1064 nm. Moreover, dust particles are of irregular shape and Mie theory is thus not
applicable for computations of their scattering properties. Also, the imaginary part of the
refractive index (RI) of dust is spectrally dependent, with a strong enhancement of the absorption
in the UV region (Patterson et al., 1977). And finally, particles in the fine and coarse mode may
have different origin, so the size dependence of the refractive index should also be considered.
The complexity of the problem outlined above demands the use    assumptions and
simplifications in the retrieval algorithms.
A widely used model for treating irregularly shaped particles is the one used in the
operational AERONET algorithm that mimics dust scattering properties with an assembly of
randomly oriented spheroids (Mishchenko et al., 1997; Dubovik et al., 2006). For typical dust
PSDs  the AERONET model provides lidar and depolarization ratios which agree reasonably
well with observed values (Wiegner et al., 2009; Esselborn et al., 2009; Tesche et al., 2009b).
The first attempts to invert lidar dust measurements into particle microphysics using the
spheroids model were recently made (Veselovskii et al., 2010; Di Girolamo et al., 2012;
Papayannis et al., 2012) but were applied to lofted layers of aged dust over Europe. The only test
of the spheroidal model relevant to pure dust was performed by using the data acquired during
the SAMUM-1 and SAMUM-2 campaigns (Müller et al., 2013). Results indicate that the
effective radii derived from lidar measurements are in reasonable agreement with the values
provided by AERONET and airplane sampling, while differences are significant for the
refractive index.
The application of spheroids to the analysis of lidar dust observations is an important step
forward when compared to the spherical particle approximation of Mie theory. Still we should
keep in mind that spheroid model was not specifically designed for lidar applications where
scattering in the backward direction is considered. For instance, as previously discussed
(Gasteiger et al., 2011; Müller et al., 2013) the spheroidal model has difficulty in reproducing
depolarization ratios ($\delta$) greater than 30%, , values that are representative for pure dust. When
using the spheroidal model, such high depolarization ratios can only be obtained when the real
($m_R$) and imaginary ($m_i$) parts of the refractive index are less than 1.5 and 0.005, respectively
(Dubovik et al., 2006), even though coincident in situ measurements of dust report higher values



(Kandler et al., 2011). To investigate these issues, more measurements near the dust origin
source and more tests of suitable inversion schemes are needed.

3       West Africa and the adjacent oceanic regions are very important locations for studying

dust properties and their influence on weather and climate. The SHADOW (Study of SaHAran
Dust Over West Africa) campaign is performing a multi-scale and multi-laboratory study of
aerosol properties and dynamics using a set of in situ and remote sensing instrumentation (multi-
wavelength Raman LIDAR, Wind-LIDAR,, nephelometer, aethalometer, sun/lunar photometer,
airborne sunphotometer, optical particle counter) in the framework of the CaPPA (Chemical and
Physical Processed in The Atmosphere) project (http://www.labex-cappa.fr/). The site is located
at IRD (Institue for Research and Development) Center, Mbour, Senegal ($14^0$N, $17^0$W). The
objective of the experiment is to report the optical, chemical and physical properties of the
aerosols as well as the source apportionment in a location where aerosol loading can be very
large and aerosol type depends on the season. Two enhanced observing periods of 7 weeks are
considered: March-April 2015 when dust originating from the Sahara/Sahel region is dominant
and December 2015-January 2016 when dust and carbonaceous aerosols resulting from fire
activities are in variable proportion and transported at different altitudes. Other types of aerosols
can also be  present such as sulfates from nearby urban areas or maritime aerosols depending on
the air mass flow. The mixed state of these various chemical components results in different
radiative properties of the aerosols.
We hereinafter focus our study on multiwavelength Mie-Raman lidar measurements
performed during the first phase of the SHADOW campaign for the period 8 March - 24 April.
During this period approximately 40 day- and night-time measurement sessions were performed
and numerous strong dust episodes were observed. Those lidar observations are used for the
analysis of the vertical distribution of the dust intensive and microphysical properties. In section
2 we describe the lidar equipment and in section 3 we provide examples of joint measurements
of wind and Raman lidars. Section 4 presents day-to-day variation of dust properties and
examples of vertical distribution of dust intensive parameters. The results of inversion of lidar
measurements into particle microphysics are given in section 5.
**2. LIDAR EXPERIMENTAL SET**



Data from three lidar systems were available during SHADOW campaign. These systems
are: aerosol micropulse lidar, wind lidar and multiwavelength Mie-Raman lidar.
*Aerosol micropulse lidar*
Cimel CE-370 micropulse lidar (www.cimel.fr) was operated 24 hours per day at 532 nm
wavelength allowing real-time monitoring of aerosol layer stratification. After correction for the
geometrical overlap factor, the lidar provides range corrected co- and cross polarized lidar
signals for heights above 300 m.
*Doppler lidar*
The wind field within the lower troposphere (<5 km) was measured by an eye safe
scanning wind lidar (Windcube WLS 100) manufactured by the LEOSPHERE company
(www.leosphere.com). This pulsed Doppler lidar operates at 1543 nm with a repetition rate of 10
kHz and uses a heterodyne technique to measure the Doppler shift of laser radiation
backscattered by aerosols. Simultaneous measurements of radial wind speed and aerosol
backscatter provides information on both  aerosol layer stratification and the dynamics of the
lower troposphere (Thobois and Soderholm, 2015). More technical details are given by (Kumer
et al., 2014; Ruchith and Ernest Raj, 2015).
During this experiment, continuous monitoring of the wind field in the range from 100 m
to 5 km with 50 m range resolution was performed. The total scanning cycle included two 180°
scans in the vertical plane along East/West and South/North axes with 1° resolution, 360°
azimuthal  scan with 2° resolution at 5° elevation angle, and line of sight (LOS) profiles at 75°
elevation in the four cardinal directions. The duration of the total cycle was approximately 10
minutes. The combination of LOS sequences is used in order to determine the three components
of the wind vector vertical profile relying on the Doppler Beam Swinging (DBS) technique
(Browning and Wexler, 1968).
*Multiwavelength Mie-Raman lidar*
The LILAS  multiwavelength Mie-Raman lidar is based on a tripled Nd:YAG Spectra
Physics INDI laser with a 20 Hz repetition rate, and pulse energy of 90/100/100 mJ at
355/532/1064 nm.  The backscattered light is collected by a 40-cm aperture Newtonian
telescope, which is inclined at an angle of 47 degrees to the horizon. The outputs of the detectors
are recorded at 7.5 m range resolution using Licel transient recorders that incorporate both
analog and photon-counting electronics. The full geometrical overlap of the laser beam and the



telescope FOV is achieved at 800 m -1400 m range depending on FOV used. The system is
designed for simultaneous detection of elastic and Raman backscatter signals and thus provides
three particle backscattering and two extinction coefficients along with depolarization ratio at
532 nm  (so called $3\beta+2\alpha+1\delta$ set). For the calibration of depolarization measurements, the so
called ±45° method, (Freudenthaler et al., 2009) was used. The uncertainty of depolarization
measurements due to calibration is estimated as ±15%. Acquiring Raman backscatter at 408 nm
permits profiling the water vapor mixing ratio (WVMR) (Whiteman et al., 1992). For calibration
of the WVMR,  radiosonde launches from the Dakar airport, located ~70 km from Mbour, were
used. The large separation between the lidar and radiosonde locations prevented an accurate
calibration of the WVMR so the WVMR data were used mainly to monitor the relative change of
the water vapor content. To improve the system capability for particle extinction measurements
at 532 nm,  rotational Raman (RR) scattering was used instead of vibrational nitrogen Raman
scattering at 608 nm (Veselovskii et al., 2015). For each profile, 4000 laser pulses were
accumulated so the temporal resolution of the measurements was approximately 3 minutes.
**3. TROPOSPHERE STRATIFIACATION AND DYNAMICS**
The aerosol layer stratification over the observation site was mixed-up and difficult to
analyze. To demonstrate the advantage of the joint use of wind and Raman lidar measurements,
we provide an example of observations performed on the night of 15-16 April. The transport
paths of different stratified air masses have been studied by using back trajectories from the
NOAA HYSPLIT model (http://ready.arl.noaa.gov/HYSPLIT.php).
For period from 23:00 UTC to 7:00 UTC on 15-16 April night, the time-height sections
of the logarithmic range corrected signal (LRCS) is shown in fig.1 while fig.2 shows the
horizontal wind speed (color scale) and direction (arrow) deduced from wind lidar and the sonic
anemometer wind measurements near the ground. Back trajectories of the air masses ending in
Mbour on 16 April 2015 at 2500 m (02:00 UTC, 06:00 UTC), at 900 m (00:00 UTC) and at 700
m (06:00 UTC) are reported in Fig.3.  These figures reveal complex stratification and dynamics
of the lower troposphere on 15-16 April: we can distinguish four layers (A-D) from 100 m to a
height of approximately 3000 m. In parallel, the wind field highlights the appearance of multi-
layered wind structure mainly consisted of a northerly wind (downward arrow) prevailing near
ground, which changes to an easterly wind (leftward arrow) with height (fig. 2).





• Layer A, located between 1000 m to 3000 m (at 00:00 UTC), is associated with a small
northerly wind speed (< 5 m/s) in the lower part of the layer, and a slightly larger easterly
wind speed (> 5 m/s) above 2000 m. Layer A can be considered to be a continentalized
maritime trade (CMT) wind which is one of oceanic origin that has been progressively
altered by continental trade (CT), as follows from the back trajectories shown in fig.3.
Therefore, this layer is characterized by a mixture of maritime and continental air.
• Layer B located between 400 m - 800 m at the beginning of the study period rises
progressively up to 700 m - 1000 m by the end of the dataset. This layer is characterized by
northeasterly winds and high aerosol loading. According to the back trajectories shown in
fig. 3, this air mass was transported from a continental area (Mali) and was mainly advected
by a southeasterly continental wind (CW).
• Layer C is a nocturnal low-level jet (LLJ). The jet core height is between 250 and 400 m
with a maximum jet speed exceeding 15 m/s. The LLJ was observed throughout the night
with a thickness that progressively increased with time perhaps being the causative
mechanism for the corresponding increase in height of layer B (fig.1). The LRCS values
within layer C decrease progressively up to the end of the observation period, perhaps due to
dilution of the aerosol loading.
• Finally, layer D corresponds to the nocturnal boundary layer (NBL) characterized by high
LRCS values and by small northerly or northwesterly wind speed (< 5 m/s). The NBL top
can be deduced from the LRCS profile discontinuity (Seibert et al., 2000) and is estimated to
at a height of approximately 200-300 m during the night.
Fig.4 shows the particle extinction at 532nm (4-a), water vapor mixing ratio (4-b), lidar
ratio (4-c) and depolarization ratio (4-d) both at 532 nm for the same time-height section as in
fig. 1 and 2. The water vapor can be used as a convenient tracer to separate dry continental air
masses from oceanic air masses that are characterized by higher vapor content. Due to the
geometrical overlap factor, the LILAS minimum height of the measurements shown in fig 4 is
800 m. Still, layer B (CW) is well observed starting at 03:00 UTC (fig.4a) due to the increase of
the layer height. The particle extinction $\alpha_{532}$ in layer A increases after 03:00 UTC while the
mixing ratio is decreasing (fig.4b). This may indicate that continental air mass advected by CT
has become dominant. The lidar ratio $LR_{532}$ of the particles associated with CT is about 55 sr





while for CMT as observed during the first part of the observation period, it is lower. The
depolarization ratio $\delta_{532}$ is about 30% in layer A and shows a small enhancement up to 35% for
layer B.
**4. DUST PARTICLE PROPERTIES DERIVED FROM RAMAN LIDAR**
**OBSERVATIONS**
*4.1. Day-to- day variation of particle intensive parameters*
One of the goals of the SHADOW campaign was to study the dust particle intensive
parameters such as extinction and backscattering Angstrom exponents together with lidar and
depolarization ratios. During March-April 2015 about 40 measurement sessions, including both
day and night time periods, were performed. In the analysis presented below only night time
measurements are considered, and for every session all lidar signals measured during the night
are temporally averaged. Moreover, for an evaluation of day-to-day variations of the particle
parameters we use only extinction and backscattering coefficients averaged within 1500 – 2000
m height layer, where a high dust concentration is frequently observed. In addition, only
observations with particle depolarization above 20% are selected to guarantee major dust
contribution.
To give an overview of the variation in aerosol loading, the aerosol optical thickness
(AOT) at 440 nm together with the extinction Angstrom exponent (EAE) $A_{380/500}^{\alpha}$ measured with
Cimel sun photometer is reported in Fig.5 for the 10 March-23 April 2015 period. The AOT was
relatively low (mainly below 0.4) for 17-28 March, but increased after 28 March reaching values
up to 2.0. The high AOTs are associated with low values of the extinction Angstrom exponent
indicating numerous dust episodes. Fig.6 shows the particle extinction $\alpha_{532}$ together with
extinction (EAE) and backscattering (BAE) Angstrom exponents $A_{355/532}^{\alpha}$, $A_{355/532}^{\beta}$ derived from
the lidar measurements for the same time period. During 28 March – 15 April several strong dust
episodes occurred as indicated by averaged over night particle extinction values as high as 0.5
km$^{-1}$. The insert in fig.6 provides the frequency distribution of observed EAE and BAE values.
Typically EAE varies in 0-0.3 range, but during dust episodes the values of EAE became
negative, decreasing to ~-0.15. The BAE averaged over night presents stronger variation,





because it is more sensitive to the change of complex refractive index (CRI) and decreases to as
low a value as -0.55 during dust events.
The day-to-day variation of the lidar ratios at 355 nm and 532 nm together with particle
depolarization ratio at 532 nm is shown in fig.7. The lidar ratios at both wavelengths vary in the
40-65 sr range and the frequency distribution for the ratio $LR_{355}/LR_{532}$ is given by insert in fig.7.
In 60% of the cases the ratio $LR_{355}/LR_{532}$ is close to 1, but during dust events this ratio increased
up to 1.4. The mean values of lidar ratios are close: $LR_{355}=54\pm8$ sr and  $LR_{532}=53\pm8$ sr. The
mean value of particle depolarization ratio is $30\pm4.5\%$, however during the dust events
depolarization ratio could increase up to $35\pm5\%$.
*4.2. Vertical distribution of particle intensive properties*
The vertical distribution of particle intensive properties is strongly influenced by the
origin of the air masses which during the SHADOW measurement period were coming either
from ocean or continental regions. In this section, we present the results for three days (13, 29
March and 10 April) characterized by different types of air masses.
*13 March*
As follows from fig.8, on 13 March at 21:00 UTC the air masses at the 3 heights (1500,
2500 and 3500m) were transported mainly over the ocean, but the back trajectory at 1500 m
presents a "loop" over continent, so the corresponding air masses may contain more dust
compared to other heights. Fig.9 shows the vertical profiles of 3β+2α measurements together
with lidar ratios $LR_{355}$, $LR_{532}$, depolarization ratio $\delta_{532}$, and Angstrom exponents $A^{\alpha}_{355/532}$ , $A^{\beta}_{355/532}$
on 13 March 2015 averaged over the 20:30-21:30 time period. The aerosol layer extended up to
3500 m but the extinction coefficient α was relatively small; at both 355 and 532 nm
wavelengths α did not exceed 0.16 km$^{-1}$. The particle depolarization ratio at 532 nm was
approximately $31\pm4.5\%$ inside the dust layer (up to ~2750 m ) and decreased to less than 15% at
3250 m. Likewise, the  $A^{\alpha}_{355/532}$  and $A^{\beta}_{355/532}$  are close to zero up to 2750 m, but start to increase
above indicating the presence of smaller particles. The lidar ratios $LR_{355}$ and $LR_{532}$ are
approximately $53\pm8$ sr inside the dust layer. Above 2750 m the values of LR are more noisy but
do not seem to change.
*29 March*



The backtrajectories from the night of 29-30 March associated to a strong dust case are
shown in Fig.10. The air masses at low altitude were transported over the continent and were
strongly loaded with dust. Fig.11 presents the vertical profiles of the same particle parameters as
in fig.10 but for 29 March. The extinction coefficient α inside the dust layer (below 1500 m) is
greater than 0.6 km$^{-1}$ for both wavelengths. The backscattering coefficient $\beta_{355}$ nm inside the dust
layer is lower than $\beta_{532}$ which is consistent with the lidar ratio R larger at 355 nm than that at 532
nm with values as large as 65 sr. The $A^{\beta}_{355/532}$ (BAE) is negative and gets near -0.8, while EAE is
still close to 0 as observed on 13 March (Fig.9). The negative values of BAE can result from the
spectral dependence of the imaginary part of the dust refractive index (RI) which is larger at 355
than at 532 nm (e.g. Patterson et al., 1977; Ansmann et al., 2011).

11       The ground based measurements performed during the SAMUM campaign demonstrated

that the imaginary part of the dust RI could vary from $m_I$=0.005 at 532 nm to $m_I$=0.02 at 355 nm
(Ansmann, et al., 2011). Such a strong enhancement of $m_I$ may lead to a decrease of the
backscattering coefficient (Veselovskii et al., 2010). To estimate the impact of the $m_I$
enhancement at 355 nm on the values of EAE and BAE at 355/532 nm wavelengths, numerical
simulations were performed. Extinction and backscattering Ångström exponents were calculated
using the model of randomly oriented spheroids as described in (Veselovskii et al., 2010) for a
bimodal particle size distribution:

$$\frac{dn(r)}{d\ln(r)} = \sum_{i=f,c} \frac{N_i}{(2\pi)^{1/2}\ln\sigma_i} \exp\left[-\frac{(\ln r - \ln r_i)^2}{2(\ln\sigma_i)^2}\right]. \qquad (1)$$

where $N_{f,c}$ is particle number density in the fine (f) and the coarse (c) mode. Each mode is
represented by a lognormal distribution with modal radius $r_{f,c}$ and dispersion $\ln\sigma_{f,c}$. For the
fine mode, values of $r_f$=0.1 μm and $\ln\sigma_f$ =0.4 were used. For the coarse mode $r_c$=1.0 μm and
three values $\ln\sigma_c$ =0.4, 0.5, 0.6 were considered. The three size distributions expressed in
volume are reported in the insert of fig.12. The ratio $N_c/N_f$ in all cases was 0.01, and the real part
of CRI was 1.55 for all wavelengths. The imaginary part was fixed at 0.005 for 532 nm while it
varied within the 0.005 – 0.05 range at 355 nm. Values of EAE and BAE as a function of $m_I$ at
355 nm are given by fig.12. The EAE shows no significant sensitivity to changes in $m_I$, but BAE



decreases rapidly as a function of $m_I$ at 355 nm. The present sensitivity study is limited but
illustrates the importance of accounting for the right spectral dependence of $m_I(\lambda)$.
***10 April***
On April 10, the air masses were coming from continental regions and particle
parameters showed large variation with height. We selected measurements during the period
0:00-2:00 UTC for which the backward trajectories at 1:00 UTC are shown in fig.13. The air
masses at 2000 m and 3000 m originate from the dust-laden continental region (Barren or
sparsely vegetated areas), while at 4500 m the air masses come from regions covered by grass
lands and savannas. Fig.14 shows profiles of the 3β+2α measurements together with particle
intensive parameters. The particle extinction increases with height reaching a maximum value of
around 0.2 km$^{-1}$ for both wavelengths at a height of 3000 m and then decreases up to 5 000m.
The EAE is approximately zero up to 3000 m and then it increases to 1.0 at 4500 m. The BAE
below 3000 m is smaller with minimum value $A^{\beta}_{355/532} \approx$ -0.5, but increase up to 4500 m where
EAE and BAE are approximately equivalent. The depolarization ratio is around 30% in the
2000-3500 m range, and decreases for higher altitudes. So we can identify different aerosol
layers with different properties: mostly pure dust layer within the 2000-3500 m altitude range
and mixed aerosols above it.
The relative humidity on 10 April was higher than on 13, 29 March, which could impact
the particle properties. Fig.15 shows the estimated profile of water vapor mixing ratio (WVMR)
obtained from the lidar measurements. WVMR is less than 3 g/kg within the dust layer and
increases above 3500 m reaching approximately 5.5 g/kg at 4000 m. The WVMR and the
relative humidity measured in Dakar at 0:00h using a radiosounding is reported on Fig.15 for
comparison. Both WVMR's measured by sounding and lidar are in agreement between 3000m
and 5000m (note that there is no sounding data between 4620 m and 3880 m). There are clearly
two distinct layers. If the derived properties of aerosols within the lower layer are representative
of dust, the air mass above 4000m brings another particle type. Particles, characterized by lower
depolarization ratio, are smaller since the EAE is increasing, and the layer is more humid since
the RH is increasing. Based on the analysis of the satellite data quick-looks (see for instance
http://earthobservatory.nasa.gov/GlobalMaps/), the back-trajectories reporting in Fig. 13 show
that the air mass at 4500m is coming from regions where fires were active during several days,





which can result in emission of smoke particles transported over M'Bour few days later. The
derived properties of aerosols within the 4000-5000m layer are consistent with this hypothesis;
the assumption of the air-mass origin is also consistent with the RH increase.
**5. INVERSION OF RAMAN LIDAR OBSERVATIONS TO THE PARTICLE**
**MICROPHYSICS**

7        The lidar $3\beta+2\alpha$ and $3\beta+2\alpha+1\delta$ observations analyzed in the previous sections can be

inverted into microphysical properties using regularization algorithm. As previously mentioned,
in the case of irregularly shaped dust particles such inversion is more complicated compared to
other aerosol types that may be well handled by spherical particle assumptions. In an earlier
study, a model of randomly oriented spheroids for dust was used (Veselovskii et al., 2010). This
model handles the dust particles as a mixture of spheres and spheroids, so an additional unknown
parameter, spheroids volume fraction (SVF), appears. The SVF in principle can be determined in
the process of inversion of $3\beta+2\alpha+1\delta$ measurements thanks to the use of depolarization ratio as
input parameter. However, for the dust layers, ina first guess, we assume a value of SVF=100%
to decrease the number of retrieved parameters. In the process of inversion we used the "search
space" parameters similar to those described in (Müller et al., 2013). The boundary of the
inversion window has been set to minimum and maximum particle radii of 0.075 and 15 μm,
respectively. The real part of RI was allowed to vary in the range 1.35 - 1.65, while the
imaginary part varied in the range 0 - 0.02. The refractive index was assumed to be spectrally
independent. The effects of a possible spectral dependence of the imaginary part of RI will be
considered at the end of this section.

23        Fig. 16 shows the particle volume density retrieved from $3\beta+2\alpha$ measurements on 13

March, 29 March and 10 April, which were discussed in the 4.2 section. The profiles of particle
volume are given together with corresponding extinction coefficients at 532 nm. The volume –
extinction ratio $V/\alpha_{532}$ for these days is also reported as an insert. Inside the dust layer this ratio
varies within the range $(0.50\text{-}0.65)*10^{-6}$ m, while outside the dust layer, the $V/\alpha_{532}$ ratio
decreases. An overview of observed values of the volume – extinction ratio for dust, obtained
from in situ, AERONET and lidar measurements is presented in Ansmann et al., 2012 and
provides $V/\alpha_{532}$ varying within the range $(0.60\text{-}1.29)*10^{-6}$ m. Thus our results are near the low
boundary of these previously published results.





The profiles of the effective radius and the real part of RI are shown in fig.17. The
inverted effective radius inside the dust layer is between 1.05 and 1.25μm (1.15±0.3 μm) and
similar for the 3 days. The AERONET retrievals provided column integrated values that are in
the same range and agrees within the uncertainty. On 30 March early morning, when the dust
contribution to the AOT is prevailing, the effective radius $r_{eff}$=1.36 μm  and it varies between
0.918 and 1.70 depending on the days and time. The lidar retrievals indicate that the real part of
the CRI in the dust layer varied from 1.51±0.05 to 1.57±0.05, which is quite typical for desert
dust (Patterson et al., 1977), while the AERONET retrievals yield values between approximately
1.46 and 1.58 depending on the days. Outside of the dust layer the retrieval of $m_R$ is not reliable
because the assumption of SVF=100% is not fulfilled and, as a result, the retrieved values of $m_R$
are overestimated (Veselovskii et al., 2010).
The values of the imaginary part of the CRI  retrieved from lidar measurements are
approximately 0.007 inside the dust layer. However, the retrieved value is unreliable since
associated to high uncertainties (Müller  et al., 2013) and, in addition, influenced by the
assumption of a spectrally independent value.
The regularization approach provides the main features of the particle volume size
distribution (PSD). Fig.18 shows the PSDs derived from lidar measurements on 10 April for four
height layers of 150 m width centered at 1940, 3150, 4070, 4370 m heights. For the layers with
strong dust loadings (1940, 3150 m) the coarse mode is dominant, at higher altitude outside the
dust layer (4070, 4370 m), the fine mode (around 0.15μm) prevails. For comparison, the column
integrated PSD obtained from AERONET level 1.5 data on 9 April at 18:00 UTC is also
reported. The coarse mode looks shifted toward larger particles when compared to the lidar
retrievals but the difference can be due to the spectral dependence of the imaginary part of $m_I$, as
it will be discussed further in this section.
Depolarization measurements provide additional information about particle properties
that can be used in the inversion algorithm as long as the forward model can compute the particle
depolarization ratio with sufficient accuracy (Veselovskii et al., 2010; Müller  et al., 2013).
Hereinafter, we compare the retrieved aerosol parameters using 3β+2α  or  3β+2α+1δ
observations. To perform such a comparison we calculated the ratio of the effective radii ( $R_{\delta}^{r}$ )



derived from 3β+2α+1δ and 3β+2α sets. Fig.19 shows the profiles of $R_\delta^r$ for the same three days
(right part associated with bottom x-axis); a value of 1.0 would mean that the additional input
has no impact on the retrieval. Inside the dust layer the ratio is about 1.15 for the measurements
taken on 13 and 29 of March. On 10 April, the ratio is noisier and more oscillating, but the
average is still close to the results obtained for 13 and 29 March. Let us mention that the ratio of
the particle volumes $R_\delta^V$ is very close to $R_\delta^r$, so it is not shown in the figure. The increase of the
effective radius (and volume) retrieved from 3β+2α+1δ measurements compared to 3β+2α
occurs simultaneously with a decrease of the real and imaginary parts of CRI (Veselovskii et al.,
2010; Müller et al., 2013). $m_R$ and $m_I$ decrease to values less than 1.45 and 0.005, respectively
and are lower than expected based on in situ measurements (Müller et al., 2013). It may suggest
that the spheroidal model has difficulty to reproduce high depolarization measurements. On
March 13 and April 10, the depolarization ratio δ is decreasing above 2500m and 3700m (Figs 9
and 14 respectively) and we can notice that the value $R_\delta^r$ is then close to 1. Assuming that results
obtained using 3β+2α data are more representative of the actual values, it means that the
spheroidal model cannot reproduce high depolarization ratios reasonably well. Additional
information brought by the depolarization ratio is so not suitable in such conditions.
The inversion results presented in fig.16, 17 are obtained assuming a spectrally
independent refractive index while the imaginary part of CRI of dust is expected to increase in
the UV spectral region. To test the effect of a spectrally dependent imaginary part $m_I(\lambda)$ on the
retrieval, we now assume that the imaginary parts at 1064 nm and 532 nm wavelengths are the
same $m_I(532) = m_I(1064)$, while $m_I(355) = 4m_I(532)$. Such an enhancement of $m_I$ at 355 nm is
quite typical for Saharan dust (Ansmann et al., 2011). The 3β+2α measurements for the same
three days are so inverted assuming this $m_I(\lambda)$ spectral dependence as described in (Veselovskii
et al., 2010). Fig.19 (left part associated with top x-axis) shows profiles of $R_{mI}^r$, which is the
ratio of the effective radii retrieved under the assumption of spectrally dependent and spectrally
independent imaginary part of RI. Again, the corresponding ratios $R_{mI}^V$ for the volumes are close
to $R_{mI}^r$ and we do not report them. As expected, the effect of $m_I(\lambda)$ is more pronounced inside the
dust layer, and on 29 March and 10 April (days characterized by negative BAE), the value of
$R_{mI}^r$ is up to 1.5. These model computations demonstrate that accounting for the spectral



1 dependence of the imaginary part of RI in the dust layers may significantly increase the retrieved

2 values of the effective radius and particle volume. In particular, it may explain smaller radii of

3 the coarse mode particles retrieved from lidar measurements inside the dust layer (fig.19) when

4 compared to AERONET results.

6 **CONCLUSION**

7  The lidar measurements performed in March-April 2015 during the first phase of the

8 SHADOW campaign in Senegal has provided a significant amount of information about dust

9 particle parameters. The use of rotational Raman scattering in the LILAS for 532 nm

10 observations improved the $\alpha_{532}$ measurements and, as a result, the calculation of lidar ratio and

11 extinction Angstrom exponent were improved as well. The mean values of lidar ratios of pure

12 dust were about 53±8 sr for both 532 nm and 355 nm wavelengths, which agrees with the values

13 observed during SAMUM 1 (Morocco) and SAMUM 2 (Cape Verde) campaigns. The mean

14 value of particle depolarization ratio at 532 nm was 30±4.5%, however during strong dust

15 episodes this ratio increased up to 35±5%, which is also in agreement with the results of

16 SAMUM campaigns. The backscattering Angstrom exponent at 355/532 nm wavelengths during

17 the dust episodes decreased to ~-0.7, while the EAE values, though being negative, were higher

18 than -0.2. Low values of BAE may be a result of enhanced dust absorption at 355 nm.

19  The inversion of 3β+2α measurements to particle microphysics in the case of dust is more

20 challenging than other types of aerosols. The use of pure dust observations somehow simplifies

21 this task, because the contribution of the particles in the fine mode (which may have different

22 origin) is insignificant. Moreover, in the retrieval of pure dust properties we don't need to

23 consider the mixture of spheres and spheroids and can assume SVF=100%. The use of the

24 spheroids model for the inversion of 3β+2α measurements leads to values of effective radius in

25 reasonable agreement with AERONET observations and yields reasonable values of the real part

26 of RI. However, the use of depolarization measurements (3β+2α+1δ) in the inversion for pure

27 dust, which is characterized by a depolarization ratio $\delta_{532}$ exceeding 30%, leads to values of

28 effective radius and volume exceeding the corresponding values obtained from 3β+2α

29 measurements by a factor up to 1.15. At the same time, the values of $m_R$ are decreased. These

30 issues have already been discussed so at the current time we prefer to not use the depolarization

31 ratio in the input data set for the inversion of dust particle parameters. On the other hand, for





particles with depolarization ratios of less than 30% the results obtained from 3β+2α and
3β+2α+1δ observations are in reasonable agreement and the use of the 3β+2α+1δ dataset in the
inversion of low depolarizing aerosols permits spheroids volume fraction to be estimated.

4        The analysis performed here also demonstrates the importance of the spectral dependence

of the imaginary part of RI in the UV spectral region. Model simulations demonstrate that
including $m_I(\lambda)$ dependence may increase the values of effective radius and volume by a factor
as large as 1.5. Thus, at the moment, dust particle microphysical properties obtained by inversion
of lidar measurements may contain significant biases. Further research is needed to develop
techniques correcting these biases in order the uncertainty of the estimates of $r_{eff}$ and V to remain
below 30%, which is a typical value when particles with prevailing fine mode are considered.

11       In addition to aerosol properties, the LILAS system provided profiles of the water vapor

mixing ratio, which, being a conserved quantity, was frequently a convenient tracer that
indicated the boundary between dry air masses transported over the continent and moist air
masses transported over the ocean. The episodes considered in this paper were characterized
mainly by low values of RH and the effects of the particles hygroscopic growth were not
considered. Still, we have significant amount of the measurements in the condition of high RH,
accompanied by formation of water and ice clouds near the dust layers. We plan to present these
results in a separate publication.
**Acknowledgments:** The authors are very grateful to IRD-Dakar (Institut de Recherche pour le
Développement) for their welcome and efficient support and also thank the labex CaPPA for
supporting this campaign. The CaPPA project (Chemical and Physical Properties of the
Atmosphere) is funded by the French National Research Agency (ANR) through the PIA
(Programme d'Investissement d'Avenir) under contract "ANR-11-LABX-0005-01" and by the
Regional Council " Nord-Pas de Calais » and the "European Funds for Regional Economic
Development (FEDER)



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





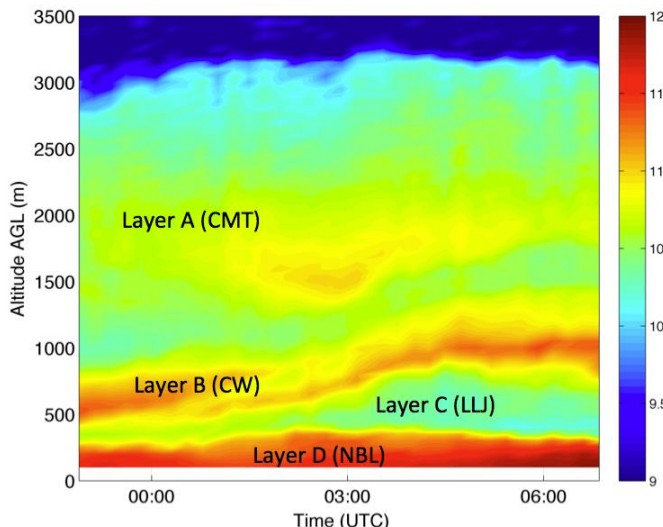

Fig.1. Time-height section of the logarithmic range corrected lidar signal deduced from the
Doppler lidar measurements during the 15-16 April night at Mbour. The stratification is
represented by four layers: (A) continentalized maritime trade (CMT), (B) Layer advected
mainly by a continental wind (CW), (C) low-level jet (LLJ) and (D) nocturnal boundary layer
(NBL).



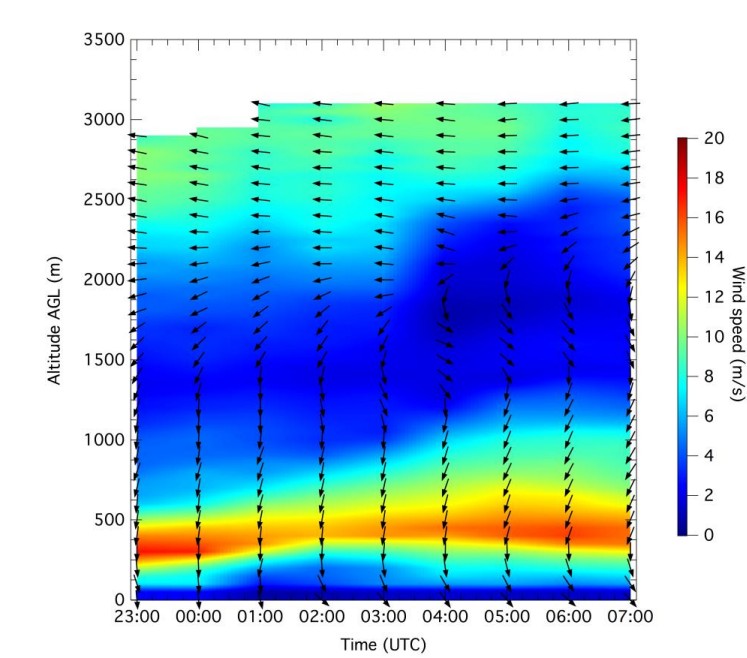


Fig. 2. Time-height section of wind direction (arrows) and wind speed (color map) deduced from
Doppler lidar during 15-16 April. Leftward and downward arrows represent, respectively,
easterly wind and northerly wind



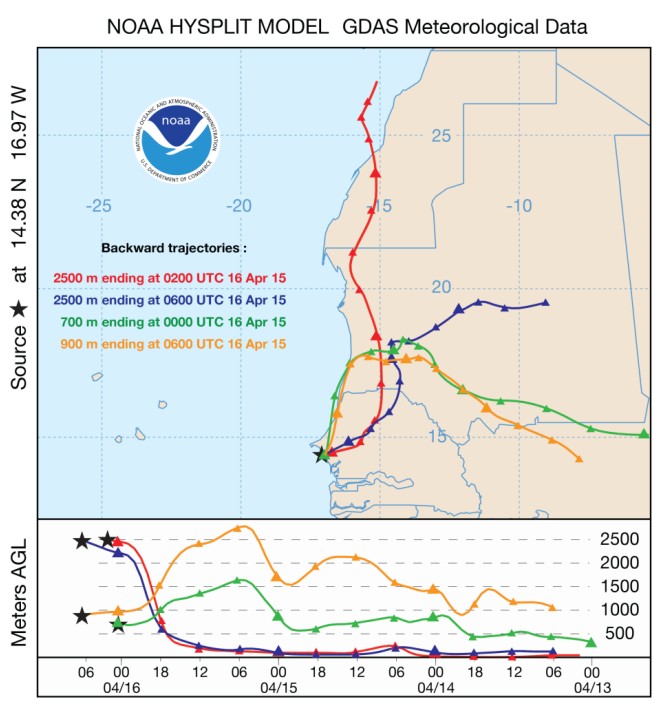

Fig.3. Back trajectories of the air masses ending in Mbour on 16 April 2015 at 2500 m (02:00
UTC, 06:00 UTC), at 900 m (00:00 UTC) and at 700 m (06:00 UTC).





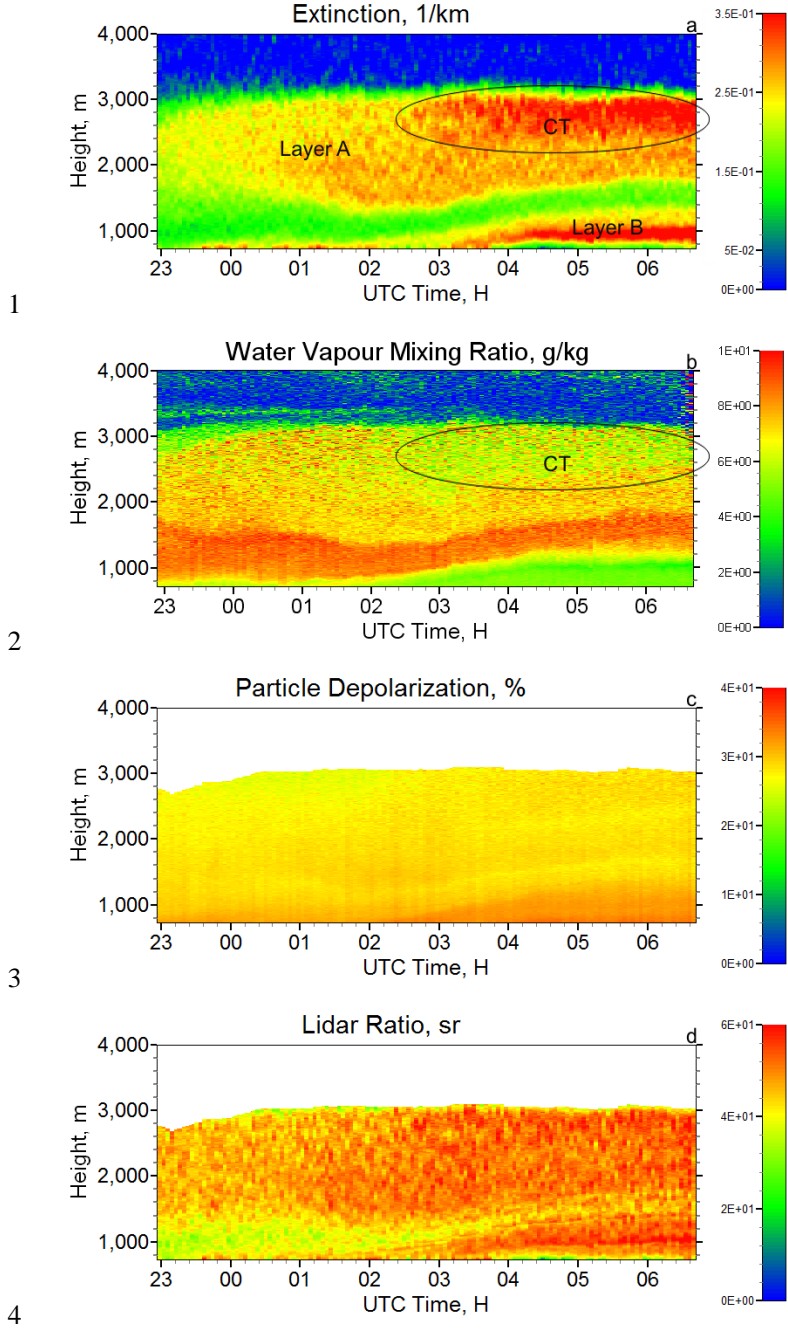

Fig.4. Height-temporal distribution of particle characteristics: (a) extinction $\alpha_{532}$, (b) water vapor
mixing ratio, (c) particle depolarization and (d) lidar ratio $R_{532}$ measured during the 15-16 April
night.



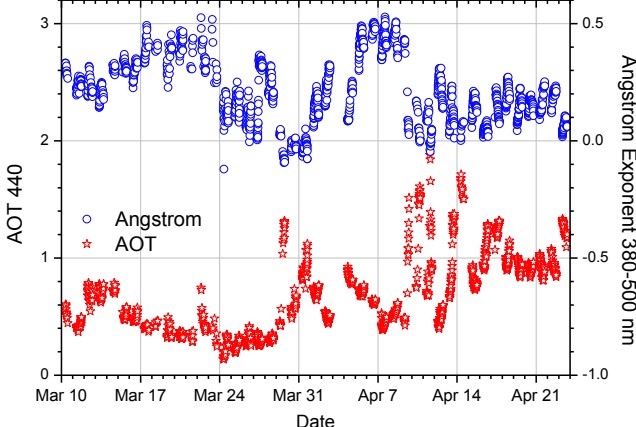

4   Fig.5.  Aerosol optical thickness (AOT) at 440 nm and the extinction Ångström exponent at 380-

5   550 nm wavelengths provided by AERONET in Mbour for March – April 2015 period.





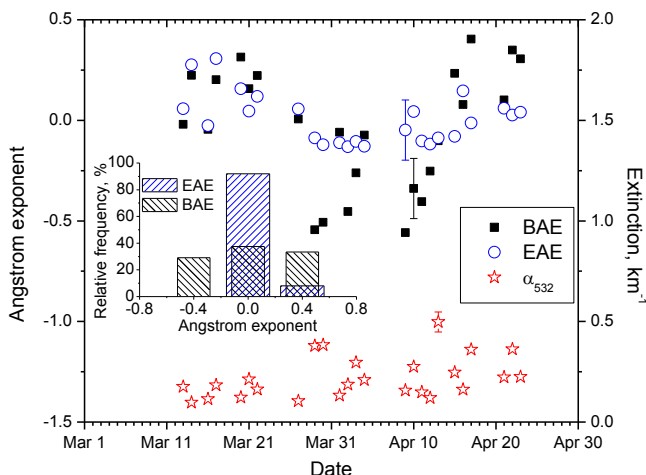

Fig. 6. Particle extinction at 532 nm together with backscattering and extinction Ångström
exponents derived from lidar measurements within 1500 m – 2000 m layer  for period March-
April 2015. The insert shows the frequency distributions of BAE and EAE.

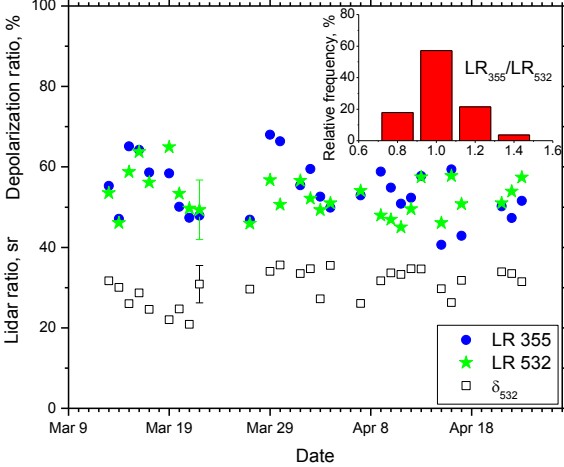

Fig. 7. Lidar ratios at 355 nm and 532 nm together with particle depolarization ratios derived
from lidar measurements within 1500 m – 2000 m layer for period March-April 2015. The insert
shows the frequency distribution of the ratio $LR_{355}/LR_{532}$.



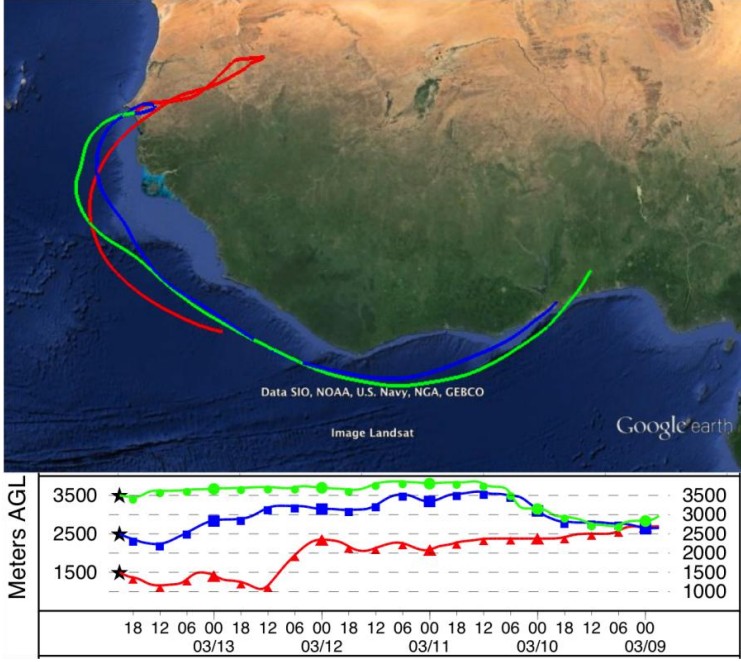

5  Fig.8. Five-day backward trajectories for the air mass in Mbour at altitudes 1500 m, 2500 m,

6  3500 m, on 13 March 2015 at 21:00 UTC.





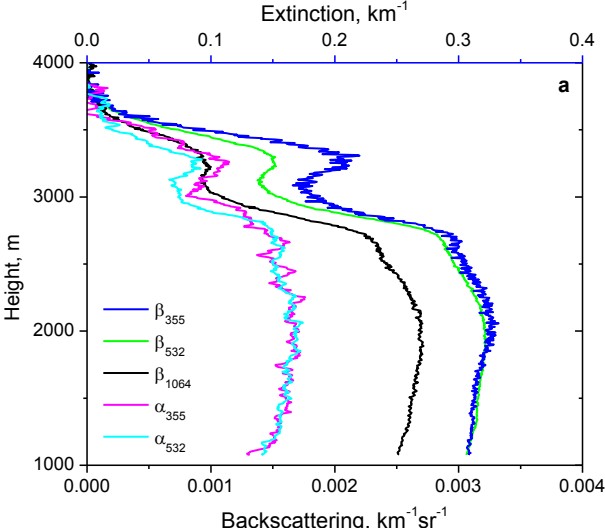

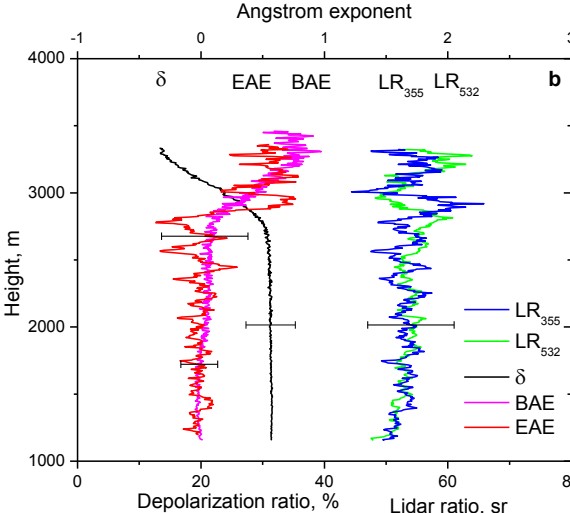

Fig.9. Vertical profiles of (a) backscattering and extinction coefficients and (b) lidar ratios,

depolarization ratio, backscattering and extinction Ångström exponents at 355/532 nm measured

on 13 March 2015 for period 20:30-21:30 UTC.





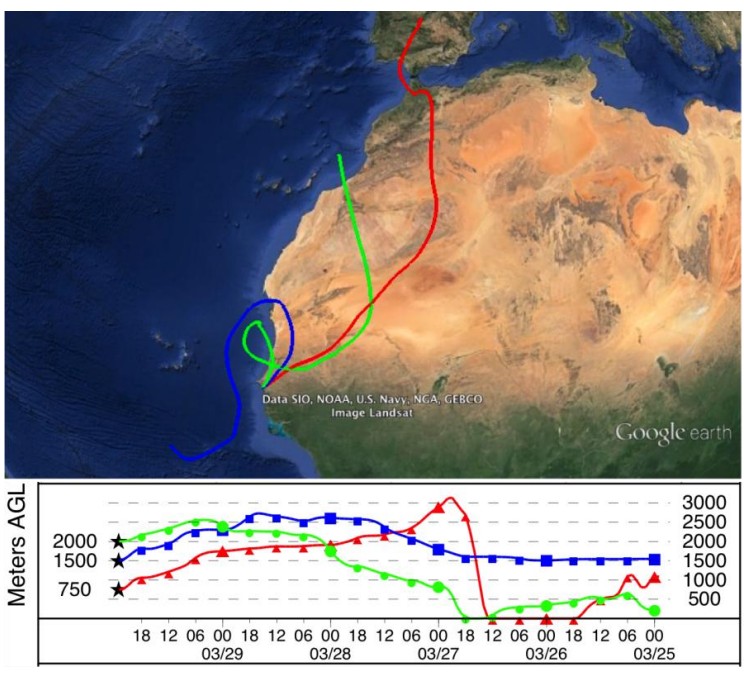

3    Fig.10. Five-day backward trajectories for the air mass in Mbour at altitudes 750 m, 1500 m,

4    2000 m on 29 March 2015 at 23:00 UTC.





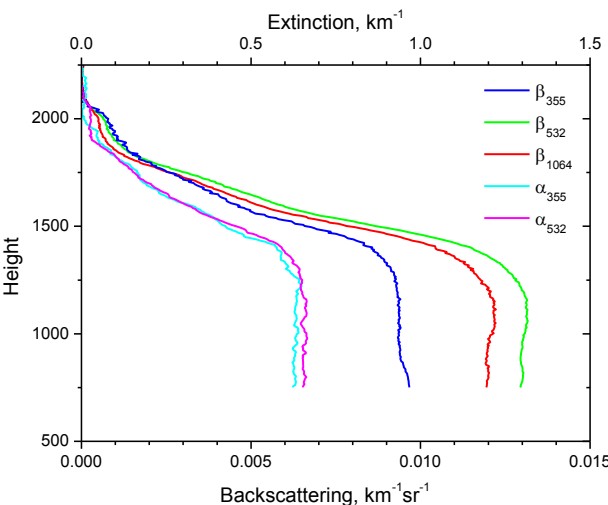

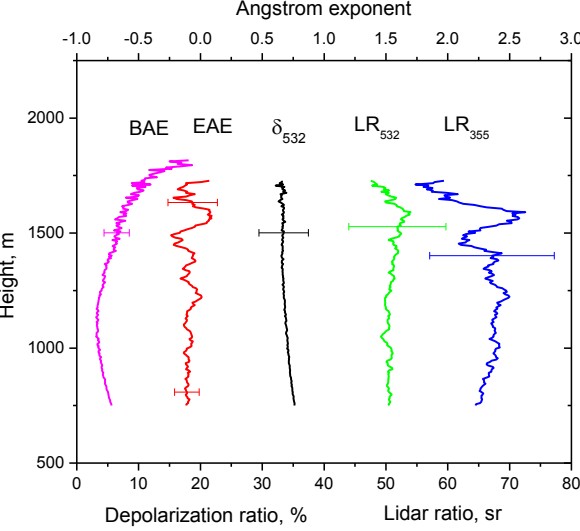

4   Fig.11. Vertical profiles of (a) backscattering and extinction coefficients and (b) lidar ratios,

5   depolarization ratio, backscattering and extinction Ångström exponents measured on 29 March

6   2015 for period 22:00-23:30 UTC.




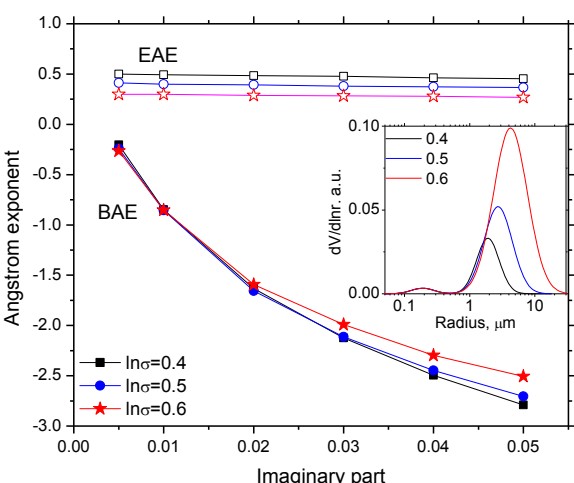

Fig.12. Extinction and backscattering Ångström exponent for 355/532nm wavelengths as a
function of the imaginary part of the refractive index at 355 nm. The CRI at 532 nm was kept
m=1.55-i.005. Computations were performed using the model of randomly oriented spheroids
for three bimodal PSDs shown in the insert.





4    Fig.13. Five-day backward trajectories for the air mass in Mbour at altitudes 2000 m, 3000 m,

5    4500 m on 10 April 2015 at 01:00 UTC.



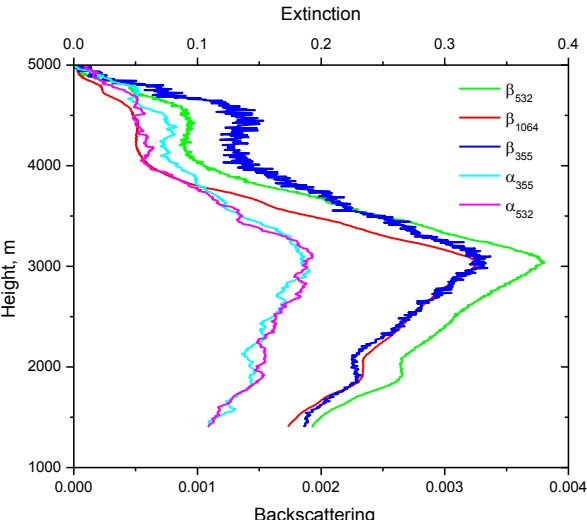

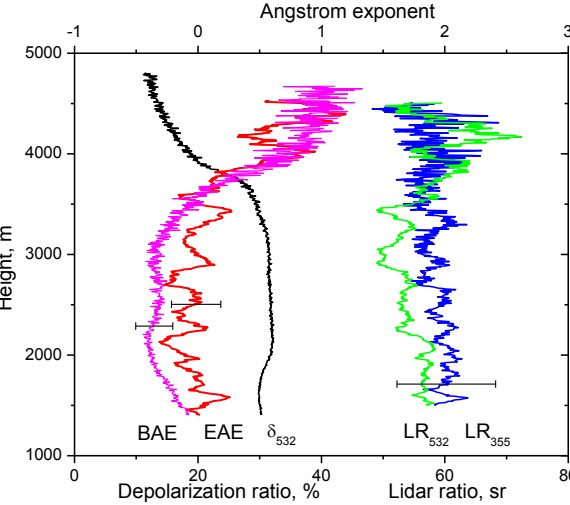

Fig.14 Vertical profiles of (a) backscattering and extinction coefficients and (b) depolarization
ratio, backscattering and extinction Ångström exponents measured on 10 April 2015 for period
00:00-02:00 UTC. Open symbols show the relative humidity and WVMR from midnight
radiosond measurements in Dakar.





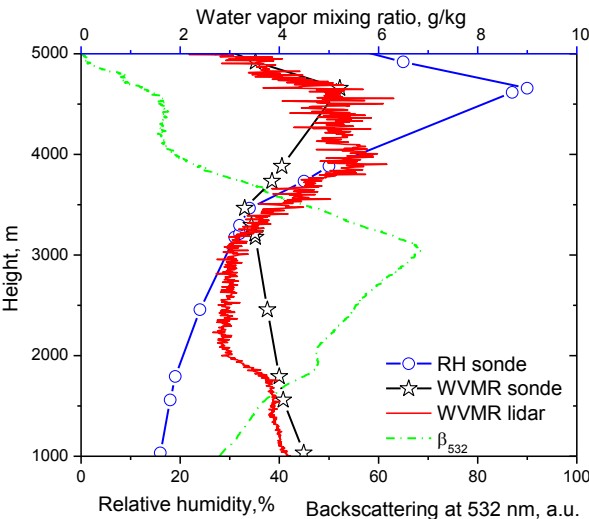

2  Fig.15. Vertical profile of water vapor mixing ratio (WVMR) measured with Raman lidar. The

3  symbols show WVMR and the relative humidity (RH) measured with radio sonde in Dakar on 10

4  April at 00:00 UTC. Green dash-dot line shows backscattering coefficient at 532 nm.





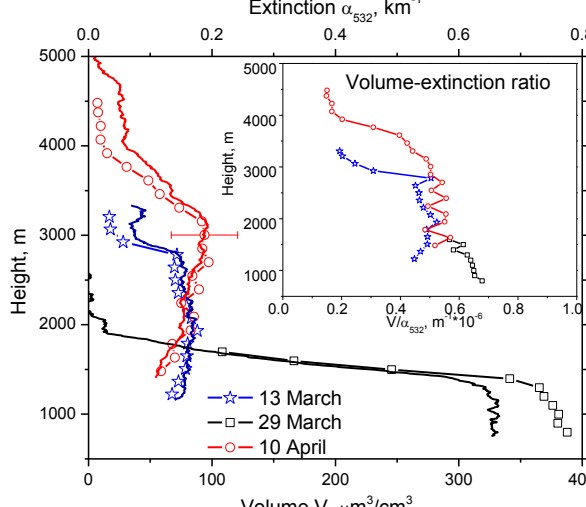

Fig.16. Vertical profiles of the particle volume density V retrieved from 3β+2α measurements on
13 March, 29 March and 10 April (symbols). Solid lines indicate the profiles of extinction
coefficient at 532 nm. The insert shows the volume – extinction ratio $V/\alpha_{532}$ for the days
considered.



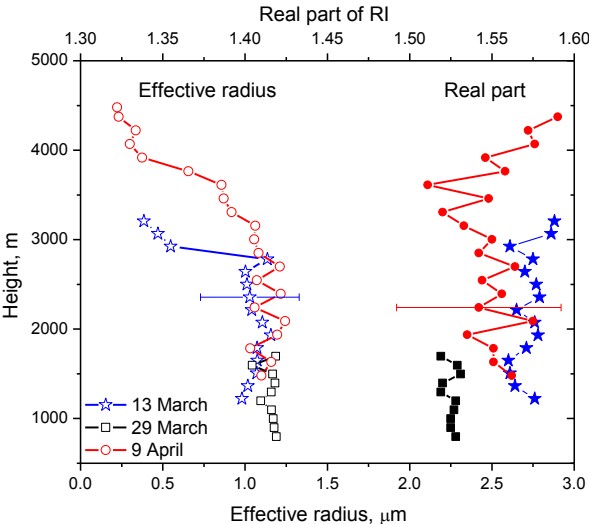

2    Fig.17. The profiles of (open symbols) the particle effective radius and (solid symbols) the real

3    part of RI retrieved from 3β+2α measurements on 13 March, 29 March and 10 April.




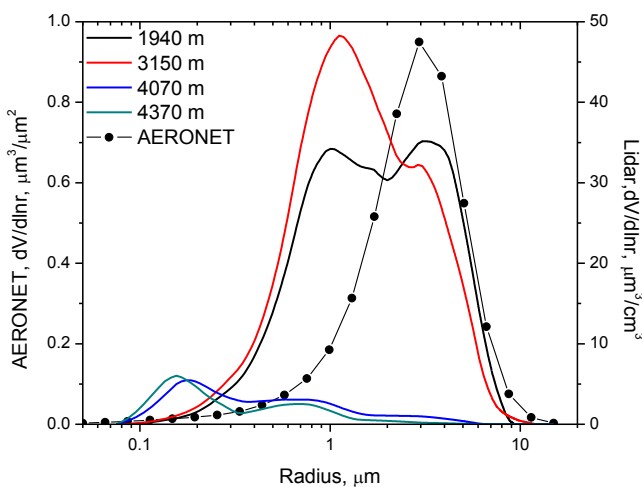

Fig.18. Particle size distributions retrieved from the measurements on 10 April for four height
layers 1940, 3150, 4070, 4370 m. Symbols show the PSD provided by AERONET on 9 April at
18:00 UTC, inversion level 1.5.

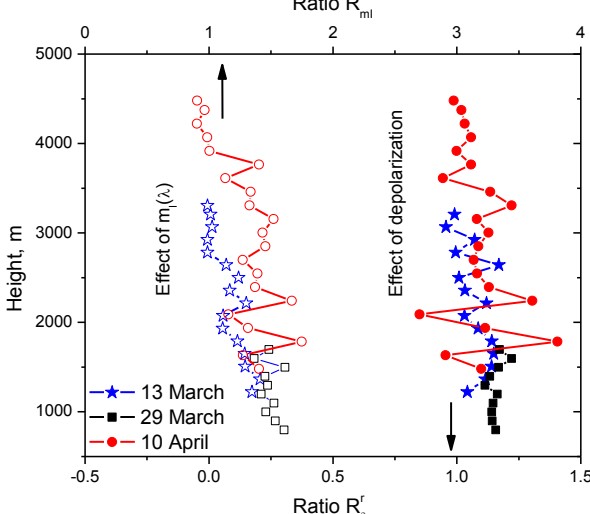

Fig. 19. Enhancement of retrieved effective radius due to using the particle depolarization ratio
in input data set ($R_\delta^r$) and due to accounting for the spectral dependence of the imaginary part of
RI ($R_{mI}^r$).Shown are results for the measurements on 13 March, 29 March and 10 April 2015.