# Peer review of "Retrieval of optical and physical properties of African dust from multi-wavelength Raman lidar measurements during the "SHADOW" campaign in Senegal"

_Atmospheric Chemistry and Physics, 2016_

## Referee Comment (RC1) · Anonymous Referee #1 · 11 Mar 2016

This paper presents case studies of multi-wavelength lidar measurements of dust including lidar ratio and angstrom exponent, adding to the published literature of case studies for dust lidar measurements. In addition, the paper applies a microphysical inversion scheme to retrieve volume concentration, effective radius and real refractive index; however, these inversions are affected by known problems with the assumptions which may affect the results. For instance, the authors use simulations to make a useful demonstration that backscatter angstrom exponent is sensitive to wavelength dependence of the imaginary refractive index, and that neglecting this spectral dependence can result in large errors in retrieved quantities. The authors also point out potential difficulties with the spheroidal model used to represent the optical properties of the

dust particles and show that this prevents the effective use of depolarization measurements in the retrieval. I find this paper to be useful given the importance of determining dust optical and microphysical properties, and in particular due to the importance of understanding the limitations of the tools currently available for determining them (and for motivating increased effort in improving those tools if possible).

However this is not the first discussion of either of these sources of retrieval bias. There's a definite conclusion that spectral dependence of the imaginary refractive index needs to be taken into account, but it's not shown how to do that in this paper. It's also not clear that the results in this study really justify the use of the spheroidal model. Instead of providing solutions to these difficulties, I believe the authors' aim is to show that some quantities can still be determined with acceptable accuracy even given these major difficulties. The biggest issue I have with this manuscript is that this premise is not clearly spelled out and is not well supported. I could be convinced that this may be true in some circumstances, but I do not find enough supporting evidence in the manuscript, or enough information to determine under what circumstances it could be true. More discussion and supporting evidence for this assertion is needed.

More information about the uncertainties in the measurments and retrievals is also needed, including error bars for all quantities at different altitudes, and also information about the vertical resolution of the measurements and retrievals.

SPECIFIC COMMENTS

Page 4, line 13. Three references are given for the statement "For typical dust PSDs the AERONET model provides lidar and depolarization ratios which agree reasonably well with observed values". I feel that these citations are misleading. The first paper, Wiegner et al. 2009, is the only one of these three that actually addresses the AERONET model. They find many limitations to the model and find that sometimes the agreement is good and sometimes quite poor. Sometimes they can improve the agreement only by shifting the refractive index away the measured values, which to my

mind calls into question whether the agreement should be considered "reasonable". I can find nothing at all about the AERONET model in the second paper, Esselborn et al. 2009. The third paper, by Tesche et al. 2009, has no new information and only quotes Wiegner et al. 2009 plus another paper which at the time was a future paper. I don't know if this future paper was published, but if so then this manuscript should probably be using that reference rather than a secondary reference.

Page 4, line 14. What kind of results did these "first attempts" find?

Page 5, line 1-2. "more measurements are needed". I certainly agree with this, but I think you have even more strongly motivated the need for improved particle modeling, both by your introduction and in the results and analysis that follow. You might consider discussing this significant need in the conclusion section.

Page 5, line 3 ff. When first encountering this paragraph I was not familiar enough with dust climatology in Africa to understand why a field mission would be designed for this time of year. It would be helpful if you'd consider adding a sentence or two about the meteorological/climatological conditions, with references, somewhere near this paragraph in order to help other readers who are unfamiliar. For example, the explanation of the Harmattan mechanism and references that were at the beginning of what was section 4.3 of the original manuscript would be fine.

Page 6, line 4 ff. Are the micropulse lidar measurements used in this paper? If not, then probably remove this section.

Page 7, line 6. Uncertainty in depolarization is estimated as plus or minus 15%. Since depolarization ratio is often expressed as a percentage, "15%" could mean either absolute or relative error. Please resolve the ambiguity.

Page 7, line 14. What is the vertical resolution of the backscatter, extinction, and depolarization ratio measurements? Not the range bin size but the effective vertical resolution: that is, how many independent measurements are in one of these profiles?

[Figure]

Section 3. I found this analysis combining the aerosol and wind measurements, plus back-trajectories, to be interesting. However, I also have some confusion about how the lidar properties reflect the characterization of these airmasses. Adding additional analysis such as was done with the other three case studies in sections 4 and 5 would really help to strengthen the logical cohesion of the paper, to better understand this observation, and also perhaps to improve the diversity of the case studies presented.

I am specifically confused about the near uniformity of the particle depolarization ratio and lidar ratio in the observations from 15-16 April. Even where the airmass is being described as "continentalized maritime" the particle depolarization ratio is in excess of 25%, perhaps implying a significant amount of dust even in this layer. Likewise, the lidar ratio appears to be quite high everywhere, much higher than would be expected for marine aerosol. Any comments about this?

Page 9, line 3 and Figures 1 and 4. It's confusing that the labels for the layers in Figure 4 don't match the labels A,B,C,D from figure 1. When "Layer B" is mentioned in the text in the discussion of figure 4, it's not clear whether this is Figure 1's layer B or Figure 4's layer B.

Page 9, line 1 "during the first part of the observation period, it is lower". Please quantify. It's difficult to read quantitative results off the color charts, so it's important to have numbers in the text.

Page 9, line 15-17, and Fig 5-7. Selections are made to guarantee major dust contribution. I think this choice to filter for only dust cases makes these figures extremely confusing. It would be much more helpful to have the full data set in the timeseries of Figures 5-7, with the observations that are dominated by dust indicated by shading or some other way. There is discussion about the variability of EAE and lidar ratio being due to dust episodes, but since only dust cases are shown, it's impossible to see the full range of variability. And it's confusing to attribute the variability within the displayed data set to dust episodes when everything shown is a dust episode. It also makes it

impossible to compare the lidar-observed values in Figures 6 and 7 with the AERONET time series in Figure 5, since Figure 5 shows the complete time-series. This mismatch undercuts the usefulness of including the AERONET timeseries. For example, there's no very clear correlation in Figure 6 between the extinction and the angstrom exponents as there is in the AERONET time series, but I'm not sure if it's really less correlated, or if it just appears that way because of the missing data. Also, Figure 6 and Figure 7 do not include the same number of points. Since both are supposed to be selected by the same criteria, this seems like a simple oversight rather than intentional, but it adds to the difficulty in comparing and interpreting the data in these figures.

Figure 6-7, continued: please clarify the error bars. One error bar is shown for each lidar time series for the whole period, except LR355 which has none. Are the uncertainties the same for every observation? What is the uncertainty for LR355? Describe the error bars in the caption. Are they random or systematic, one sigma or two, etc.?

Section 4.2. It seems that all the case studies are pure dust, and all the depolarization ratios shown in all figures are quite high, except where the extinction and backscatter are lower perhaps implying there is not much aerosol at all. It would be useful to see a contrasting case, if one exists. If there is no case with significant aerosol that is not dominated by dust, then a case with a smaller dust fraction (perhaps the case from Figure 4) would still be good to see. Without getting to see some dynamic range of the measurements (again, except where the signal is much lower), it's less convincing that the measurements and retrievals are accurate. On the other hand, it may be that there just is no other aerosol type present at that time and place at high enough concentration to make much difference to the measurements. A fuller discussion of the instrument accuracy and uncertainties, including complete error bars on the figures, would help with understanding which variability is significant.

Page 11, line 8. This is a good point, and the following demonstration is useful. But BAE is a non-monotonic function of particle radius even for spectrally constant complex refractive index and spherical particles (i.e. just from Mie modeling). Is spectral dependence of the imaginary part of the refractive index the only way to achieve significantly negative BAE?

Page 11, line 12. Does the real part have a spectral dependence as well?

Page 11, continued. The large negative BAE signature is present in this case but not the previous case. Is this discussion meant to imply that there is spectral dependence in the imaginary refractive index only in some of these cases? What is the explanation for different dust layers having strong spectral dependence of the imaginary refractive index in some cases but not others?

Page 12, line 2. I agree the study demonstrates the importance of accounting for spectral dependence of the imaginary refractive index. Is it possible to do this using values of EAE and BAE more similar to the measurements for this case, to demonstrate more conclusively that this measurement case is in a regime where this effect is significant?

Page 12, lines 10-17. The upper layer where the intensive properties are different appears to have fairly small extinction and backscatter, meaning less signal. Error bars showing the uncertainties in the upper layers would make the attribution of this variability to differences in aerosol type more convincing.

Figures 9, 11, 14, 15, 16. Again, more details about the error bars would be good. Include error bars on all quantities at more than just one altitude and describe them in the captions. The text suggests that the uncertainties vary; the figures should reflect this.

Page 12, line 18. "The relative humidity on 10 April was higher than on 13, 29 March". Can you show RH for the other two cases also? Or at least quantify the values for the earlier cases in the text.

Page 13, lines 10-11. Please specify, is this the same version of the algorithm you are using in this study?

Page 13, line 13. Consider using a different acronym. It's difficult to remember later if

"S" is spheres or spheroids. Maybe NSVF for "non-spherical volume fraction".

Page 14, line 4. It would be useful to show the AERONET comparison on the figures.

Page 14, 2-12. What is the AERONET non-spherical fraction? Does this support the use of the assumption of 100% non-spherical fraction?

Page 14, line 17 and Figure 18. "the main features of the particle volume size distribution". Can you be more specific about how much information is provided by inversion about the size distribution (how many bins or coefficients) and how it's obtained? These size distributions seem surprisingly detailed for an inversion of just 5 pieces of information. I suppose this is probably explained in an earlier paper, but a brief description of the inversion (probably in a new section between section 2 and 3) would still be helpful here in this paper.

Page 14, line 14. If the retrieved imaginary refractive index is unreliable due to errors associated with a faulty assumption in the retrieval, what evidence is there that the other retrieved variables are trustworthy? While I believe they may be, it doesn't seem obvious that this must be so. The question certainly deserves more discussion, if any of the retrieval results are to be considered useful.

Page 15, line 14. "Assuming that results obtained using 3-beta plus 2-alpha data are more representative of the actual values..." Similar to the previous comment: this seems like a big assumption and very important to the analysis of the results. If the measured depolarization ratio can't be reproduced by the spheroidal model but we want to believe the results of the inversion of backscatter and extinction only, then we need to be convinced that the spheroidal model can at least correctly determine backscatter and extinction. It seems that backscatter would be of particular concern, since, as pointed out in the introduction "the spheroid model was not specifically designed for lidar applications where scattering in the backward direction is considered." Indeed the introduction suggests that previous studies found that the spheroid model also leads to errors in refractive index. Is there additional analysis that can be done to

demonstrate the correctness of the inversion of the 3-beta plus 2-alpha data using the spheroid model or to better characterize the errors?

Page 15 and Figure 19. What is the effect of these two experiments on other retrieved quantities, like the volume concentration or the real refractive index?

Page 17, line 3. One of the conclusions is that for small enough depolarization the 3-backscatter + 2-extinction + 1-depolarization inversion permits the spherical/non-spherical fraction to be estimated. This isn't part of the analysis of the paper and there is no real support for it here; it might be better to delete it. If it is kept, then besides supporting it with further analysis, it would also be good to clarify whether this gives more complete or more accurate information than the spherical/non-spherical separation that has been practiced for lidar measurements for over a decade (Sugimoto and Lee, 2006; Tesche et al. 2009)

Figure 1, caption. What quantity is the "lidar signal"?

Figure 3, caption and annotation. It would be useful to explain the correspondence between the four trajectories and the regions A,B,C,D from Figure 1.

TECHNICAL COMMENTS

Page 16, line 20. "Somehow" = "somewhat"

Figure 4, the labels on the color axis are too small.

REFERENCES

Sugimoto, N., and C. H. Lee (2006), Characteristics of dust aerosols inferred from lidar depolarization measurements at two wavelengths, Appl Optics, 45(28), 7468-7474.

Tesche, M., A. Ansmann, D. Müller, D. Althausen, R. Engelmann, V. Freudenthaler, and S. Groß (2009), Vertically resolved separation of dust and smoke over Cape Verde using multiwavelength Raman and polarization lidars during Saharan Mineral Dust Experiment 2008, J. Geophys. Res., 114(D13), D13202.

---

## Referee Comment (RC2) · Anonymous Referee #3 · 18 Mar 2016

The paper adds information on the existing literature for desert dust properties. Dust has been measured by means of lidar remote sensors and inversion techniques were applied to derive microphysics near the source and during a field campaign in Senegal. The paper is suitable for publication in ACP but I have 2 major comments that I would urge the authors to take into account:

1. The authors conclude that the negative values of the BAEs measured for dust are due to the enhanced absorption in the UV. This is not supported by independent measurements. It is also well-known that the spheroid model assumes a spectrally independent phase function at 180 degrees. A possible spectral dependence on the 180 phase function could also be the source of negative BAEs and this limitation of the

spheroid model should be mentioned in the manuscript and in the conclusion section.

2. The paper gives the impression that the particle depolarization ratio does not provide significant information on the inversion. However, there is much discussion in the literature (see for example the work of Gasteiger) that the spheroid model cannot reproduce the lidar measurements of the linear particle depolarization ratio. Thus, how we expect that an inversion code based on the spheroid model would show that there is an added value on the microphysical retrievals by adding depolarization information? I think that the conclusions should be rephrased, such as to make clear that this could be a limitation of the spheroid model as well.

---

## Referee Comment (RC3) · Anonymous Referee #2 · 5 Apr 2016

General Comments:

Careful measurements have been performed in a region (Senegal), which is highly interesting for dust measurements. The SHADOW campaign seems very promising for further dust and dust mixture studies. The 3 case studies and the time series presented are a helpful contribution to the global community of remote measuring groups as measurements close to the Sahara are rare. A good approach to the particle microphysics is shown. So I recommend the manuscript for publication with 4 major concerns and some minor comments.

Major Comments: 1. The title: You should mention the "inversion" in the title, so it becomes more specific. Take care with the title, especially when you are planning a

second part of the "SHADOW" campaign.

2. Chapter 3 "Troposphere Stratification and Dynamics": It is not linked with rest of the manuscript and it is not mentioned in the conclusion. Maybe you should prepare a separate publication dealing with wind information; there you may use the information of the micropulse lidar, too. It was not used for this publication, although it provides depolarization information down to 300m. Or you should present one of your three case studies in this detailed manner. But in total it is still a very nice measurement case to introduce the possibilities of your instruments.

3. Your LILAS lidar: Is there a reference describing the lidar system in more detail? A well characterized lidar system is crucial for the data quality. Why you use 47°? I assume that your entire lidar system is inclined by 47°. Have you measured the transmission ratios (transmittance of cross and parallel polarized light) of your detection unit as they may affect the total signal in presence of heavy dust plumes with high depolarization ratios?

4. Comparing your results to the AERONET retrieval (p16l25) is not the final proof as AERONET is using a particle shape model, too. So AERONET is affected by non-spherical particles with high depolarization ratio, too. It would be better to compare your values to in situ measurements. So I suppose using the depolarization information from a lidar system would improve the inversion.

Minor Comments:

p4l7 "demands the use of assumptions"

p5l21 add the year to the date (2015)

p7l19 add the year to the date (2015)

p8l20-21 "to at" something is missing

p12 chap "10 April" You have not considered the lidar ratio. Please comment on it while

classifying the aerosol layers.

p13l15 "in a first guess" (missing space)

p13l18 15$\mu$m as a maximum particle radius seems small so close to the desert.

fig 1: add the year to the date (2015); add a description to the color bar;

fig 2: mention "horizontal" wind to not get confused with vertical winds

fig 4: Wouldn't you expect higher depolarization values for the more continental air (CT)? Lidar ratios for marine particles in Layer A are very high. Could you please comment on this?

Fig 6: "Typically EAE varies in 0-0.3 range, but during dust episodes the values of EAE became negative, decreasing to $\sim$-0.15." It is not seen in the insert of fig 6, as nearly all cases of EAE are in the Ångström box "0". You may choose a different scaling for the insert diagram.

Fig 7: To see the variability in the particle depolarization a separate diagram for the depolarization values would be nice. There is enough space to put error bars to every depolarization value without confusing the reader, so it would be a good idea to add the error bars.

Fig 9a: beta 1064 should be shown in red as in the other plots.

Fig 14: Are you sure with the peak of beta532 at 3 km? See description: there are no open symbols (this belongs to fig 15, I suppose).

---

## Author Comment (AC1) · 4 May 2016

*1. There's a definite conclusion that spectral dependence of the imaginary refractive index needs to be taken into account, but it's not shown how to do that in this paper.*

The approach for accounting the spectral dependence of imaginary part was described in our previous publication (Veselovskii et al., 2010). Along the retrieval process, $mI(532)$ is adjustable while the ratio $mI(355)/mI(532)$ is fixed. This ratio is taken from in situ measurements, for example, from SAMUM campaign data. Such approach to some extend allows to correct effects related to spectral dependence of mI.

The comment is added to the text.

*2. I believe the authors' aim is to show that some quantities can still be determined with acceptable accuracy even given these major difficulties. The biggest issue I have with this manuscript is that this premise is not clearly spelled out and is not well supported. I could be convinced that this may be true in some circumstances, but I do not find enough supporting evidence in the manuscript, or enough information to determine under what circumstances it could be true. More discussion and supporting evidence for this assertion is needed.*

We agree with reviewer, that retrieval of particle parameters from lidar measurements is very challenging. Although we cannot provide "supporting evidences", we believe it is possible even in the case of dust as long as the required assumptions along the inversion process are correct. We are confident in our results since

- Retrieved values of effective radius are close to the values provided by AERONET

- Volume-extinction ratio is inside the range of values provided by several instruments (Ansmann et al., 2012) using different approaches;

- Real part of Ri obtained from lidar measurements agrees with in situ probes from aircraft (Muller et al., 2013).

but we agree that we cannot retrieve all parameters without some prescribed constraints .

*3. More information about the uncertainties in the measurements and retrievals is also needed, including error bars for all quantities at different altitudes, and also information about the vertical resolution of the measurements and retrievals.*

Uncertainties are added to plots in the revised manuscript. The vertical resolution is also provided.

*4. Page 4, line 13. Three references are given for the statement "For typical dust PSDs the AERONET model provides lidar and depolarization ratios which agree reasonably well with observed values". I feel that these citations are misleading. The first paper, Wiegner et al. 2009, is the only one of these three that actually addresses the AERONET model. They find many limitations to the model and find that sometimes the agreement is good and sometimes quite poor. Sometimes they can improve the agreement only by shifting the refractive index away the measured values, which to my mind calls into question whether the agreement should be considered "reasonable". I can find nothing at all about the AERONET model in the second paper, Esselborn et al. 2009. The third paper, by Tesche et al. 2009, has no new information and only quotes Wiegner et al. 2009 plus another paper which at the time was a future paper. I don't know if this future paper was published, but if so then this manuscript should probably be using that reference rather than a secondary reference.*

The reviewer is right, the last two references are not addressing AERONET model and we decide to remove them since they are misleading. They were mentioned to confirm that our values are typical.

*5. Page 4, line 14. What kind of results did these "first attempts" find?*

Inversion of lidar measurements lead to typical values of effective radius and refractive index for dust particles.

*6. Page 5, line 1-2. "more measurements are needed". I certainly agree with this, but I think you have even more strongly motivated the need for improved particle modeling, both by your introduction and in the results and analysis that follow. You might consider discussing this significant need in the conclusion section.*

In Conclusion we have added the sentence:

"We hope also that these discussions will stimulate development of the forward model accurately describing polarization properties of laser radiation backscattered by dust particles."

*7.      Page 5, line 3. When first encountering this paragraph I was not familiar enough with dust climatology in Africa to understand why a field mission would be designed for this time of year. It would be helpful if you'd consider adding a sentence or two about the meteorological/climatological conditions, with references, somewhere near this paragraph in order to help other readers who are unfamiliar. For example, the explanation of the Harmattan mechanism and references that were at the beginning of what was section 4.3 of the original manuscript would be fine.*

We have added a sentence about Harmattan mechanism with appropriate reference.

*8.      Page 6, line 4 ff. Are the micropulse lidar measurements used in this paper? If not, then probably remove this section.*

Removed

*9.      Page 7, line 6. Uncertainty in depolarization is estimated as plus or minus 15%. Since depolarization ratio is often expressed as a percentage, "15%" could mean either absolute or relative error. Please resolve the ambiguity.*

Corrected for "relative uncertainty"

*10.      Page 7, line 14. What is the vertical resolution of the backscatter, extinction, and depolarization ratio measurements? Not the range bin size but the effective vertical resolution: that is, how many independent measurements are in one of these profiles?*

We now provide the information in the text

"The backscattering coefficients and depolarization ratio were calculated with range resolution 7.5 m (corresponding height resolution 5.5.m). Vertical resolution of extinction measurements varied with height from 50 m (at 1000 m) to 125 m at (at 7000 m)."

*11.      Section 3. I found this analysis combining the aerosol and wind measurements, plus back-trajectories, to be interesting. However, I also have some confusion about how the lidar properties reflect the characterization of these airmasses. Adding additional analysis such as was done with the other three case studies in sections 4 and 5 would really help to strengthen the logical cohesion of the paper, to better understand this observation, and also perhaps to improve*

*the diversity of the case studies presented. I am specifically confused about the near uniformity of the particle depolarization ratio and lidar ratio in the observations from 15-16 April. Even where the airmass is being described as "continentalized maritime" the particle depolarization ratio is in excess of 25%, perhaps implying a significant amount of dust even in this layer. Likewise, the lidar ratio appears to be quite high everywhere, much higher than would be expected for marine aerosol. Any comments about this?*

First part of the comment: The goal of Section 3 was to illustrate complexity of troposphere dynamic and to show capabilities of the system, so we stayed detailed of particle properties for Sections 4,5.

Second part of the comment : Yes, depolarization didn't change much for that particular day (we had days with much stronger variations), so content of dust was high even for air masses which we consider as "continentalized maritime". We have added corresponding comment to this section.

*12.    Page 9, line 3 and Figures 1 and 4. It's confusing that the labels for the layers in Figure 4 don't match the labels A,B,C,D from figure 1. When "Layer B" is mentioned in the text in the discussion of figure 4, it's not clear whether this is Figure 1's layer B or Figure 4's layer B.*

Layer B is the same for both Fig.1 and Fig.4.

*13.    Page 9, line 1 "during the first part of the observation period, it is lower". Please quantify. It's difficult to read quantitative results off the color charts, so it's important to have numbers in the text.*

We have added the value "45 sr" in the text.

*14.    Page 9, line 15-17, and Fig 5-7. Selections are made to guarantee major dust contribution. I think this choice to filter for only dust cases makes these figures extremely confusing. It would be much more helpful to have the full data set in the time series of Figures 5-7, with the observations that are dominated by dust indicated by shading or some other way. There is discussion about the variability of EAE and lidar ratio being due to dust episodes, but since only dust cases are shown, it's impossible to see the full range of variability. And it's confusing to attribute the variability within the displayed data set to dust episodes when everything shown is a dust episode. It also makes it impossible to compare the lidar-observed values in Figures 6 and 7 with the AERONET time series in Figure 5, since Figure 5 shows the complete time-series. This mismatch undercuts the usefulness of including the AERONET timeseries. For example, there's no very clear correlation in Figure 6 between the extinction and the angstrom exponents as there is in the AERONET time series, but I'm not sure if it's really less correlated, or if it just appears that way because of the missing data. Also, Figure 6 and Figure 7 do not include the same number of points. Since both are supposed to be selected by the same criteria, this seems like a simple oversight rather than intentional, but it adds to the difficulty in comparing and interpreting the data in these figures.*

Fig.6 is corrected and the missing points are added.

That days with lower depolarization were characterized also by low particle extinction so we couldn't calculate intensive parameters with sufficient accuracy. For such clean nights we didn't perform full night measurements, to save the laser life time.

The term "dust episode" we used for the days with increased dust content.

*15.	Figure 6-7, continued: please clarify the error bars. One error bar is shown for each lidar time series for the whole period, except LR355 which has none. Are the uncertainties the same for every observation? What is the uncertainty for LR355? Describe the error bars in the caption. Are they random or systematic, one sigma or two, etc.?*

For these averaged data the errors are systematical. We have added the following paragraph in the text
"For such extensive averaging the uncertainties of derived parameters are mainly due to the systematical errors. Thus we estimate the uncertainty of extinction and lidar ratio calculation to be below 10% and 15% respectively for both wavelengths. Uncertainties of extinction and backscattering Angstrom exponents derivation are estimated to be below ±0.2."

*16.	Section 4.2. It seems that all the case studies are pure dust, and all the depolarization ratios shown in all figures are quite high, except where the extinction and backscatter are lower perhaps implying there is not much aerosol at all. It would be useful to see a contrasting case, if one exists. If there is no case with significant aerosol that is not dominated by dust, then a case with a smaller dust fraction (perhaps the case from Figure 4) would still be good to see. Without getting to see some dynamic range of the measurements (again, except where the signal is much lower), it's less convincing that the measurements and retrievals are accurate. On the other hand, it may be that there just is no other aerosol type present at that time and place at high enough concentration to make much difference to the measurements. A fuller discussion of the instrument accuracy and uncertainties, including complete error bars on the figures, would help with understanding which variability is significant.*

In the phase 1 of the measurements we didn't have situations when aerosol other than dust occurred in significant amounts. Still we presented results for 9 April, where above 3750 m depolarization drops and  particles become smaller. The phase 1 of campaign was focused at study of pure dust. During phase 2 the episodes of dust-smoke mixture were detected and these results will be presented separately. We have added corresponding comment to Conclusion.

*17.	Page 11, line 8. This is a good point, and the following demonstration is useful. But BAE is a non-monotonic function of particle radius even for spectrally constant complex refractive index and spherical particles (i.e. just from Mie modeling). Is spectral dependence of the imaginary part of the refractive index the only way to achieve significantly negative BAE?*

For typical size distributions and typical  refractive indices of dust the values of BAE are positive, so it is difficult to explain the observed results without considering the spectral dependence of mI.

*18.	Page 11, line 12. Does the real part have a spectral dependence as well?*

Based on literature, the real part of dust is supposed to be spectrally independent in the spectral range we consider.

*19.	Page 11, continued. The large negative BAE signature is present in this case but not the previous case. Is this discussion meant to imply that there is spectral dependence in the imaginary refractive index only in some of these cases? What is the explanation for different dust layers having strong spectral dependence of the imaginary refractive index in some cases but not others?*

The spectral dependence of dust imaginary part is probably determined by the mineralogy at the origin of the event. In situ measurements of mI during SAMUM campaigns demonstrated strong variability of spectral dependence. So we think this is the reason.

*20.      Page 12, line 2. I agree the study demonstrates the importance of accounting for spectral dependence of the imaginary refractive index. Is it possible to do this using values of EAE and BAE more similar to the measurements for this case, to demonstrate more conclusively that this measurement case is in a regime where this effect is significant?*

Typical values of mI at 532 and 355 nm are 0.005 and 0.02 respectively, estimations show that corresponding BAE may be as low as -1.5, so the effect should be significant. Unfortunately we had no information about actual spectral dependence of mI, so it is difficult to perform more detailed analysis.

*21.      Page 12, lines 10-17. The upper layer where the intensive properties are different appears to have fairly small extinction and backscatter, meaning less signal. Error bars showing the uncertainties in the upper layers would make the attribution of this variability to differences in aerosol type more convincing.*

Error bars are added

*22.      Figures 9, 11, 14, 15, 16. Again, more details about the error bars would be good. Include error bars on all quantities at more than just one altitude and describe them in the captions. The text suggests that the uncertainties vary; the figures should reflect this.*

Error bars are added

*23.      Page 12, line 18. "The relative humidity on 10 April was higher than on 13, 29 March". Can you show RH for the other two cases also? Or at least quantify the values for the earlier cases in the text.*

On 13 March RH was below 38% in 1000 m – 2600 m height range and increased up to 75% above 3350 m. On 29 March RH was below 19% in 600 m – 1500 m height range. This information is now provided in the text.

*24.       Page 13, lines 10-11. Please specify, is this the same version of the algorithm you are using in this study?*

Yes, it is the same version. Corresponding comment is added.

*25.      Page 13, line 13. Consider using a different acronym. It's difficult to remember later if "S" is spheres or spheroids. Maybe NSVF for "non-spherical volume fraction".*

Since we provide the meaning of the acronym in the manuscript, we prefer to keep the present notation that we used in all our previous publications.

*26.      Page 14, line 4. It would be useful to show the AERONET comparison on the figures.*

Added

*27.       Page 14, 2-12. What is the AERONET non-spherical fraction? Does this support the use of the assumption of 100% non-spherical fraction?*

The AERONET site reports SVF>98%, so our assumption looks valid.

*28.     Page 14, line 17 and Figure 18. "the main features of the particle volume size distribution". Can you be more specific about how much information is provided by inversion about the size distribution (how many bins or coefficients) and how it's obtained? These size distributions seem surprisingly detailed for an inversion of just 5 pieces of information. I suppose this is probably explained in an earlier paper, but a brief description of the inversion (probably in a new section between section 2 and 3) would still be helpful here in this paper.* Size distribution is represented by superposition of five base functions that we described several times in previous publications. We could repeat it herein but we feel there is no strong arguments for doing it as long as you quote the relevant papers..

*29.     Page 14, line 14. If the retrieved imaginary refractive index is unreliable due to errors associated with a faulty assumption in the retrieval, what evidence is there that the other retrieved variables are trustworthy? While I believe they may be, it doesn't seem obvious that this must be so. The question certainly deserves more discussion, if any of the retrieval results are to be considered useful.*

This topic was discussed in a previous paper. In Veselovskii et al., (Atmos. Meas. Tech., 6, 2671–2682, 2013), we showed that the exact value of mI didn't impact significantly the retrievals of effective radius and volume density when it could impact the value of mR. The corresponding uncertainty is estimated to be as +/-0.05. We have added the corresponding comment in the text.

*30.     Page 15, line 14. "Assuming that results obtained using 3-beta plus 2-alpha data are more representative of the actual values. . ." Similar to the previous comment: this seems like a big assumption and very important to the analysis of the results. If the measured depolarization ratio can't be reproduced by the spheroidal model but we want to believe the results of the inversion of backscatter and extinction only, then we need to be convinced that the spheroidal model can at least correctly determine backscatter and extinction. It seems that backscatter would be of particular concern, since, as pointed out in the introduction "the spheroid model was not specifically designed for lidar applications where scattering in the backward direction is considered." Indeed the introduction suggests that previous studies found that the spheroid model also leads to errors in refractive index. Is there additional analysis that can be done to demonstrate the correctness of the inversion of the 3-beta plus 2-alpha data using the spheroid model or to better characterize the errors?*

We agree with reviewer, that it is an important issue. However we presently cannot estimate the level of accuracy by which the backscattering coefficients is reproduced using spheroids (we only can state that it definitely better than using spheres). Only laboratory measurements can provide corresponding numbers, but it is not a trivial task. On the other hand, we have indirect indications that spheroid model works rather well: Tthe column integrated values of volume and effective radius obtained during SAMUM and SHADOW campaigns agree reasonably with AERONET results. The volume/extinction ratio obtained from lidar is inside the range of values provided by other instruments. It's our objective to pursue the present work to respond to this question more accurately.

*31.     Page 15 and Figure 19. What is the effect of these two experiments on other retrieved quantities, like the volume concentration or the real refractive index?*

As noted in the text the effects of these two experiments on volume and effective radius are quite similar, so we show results for radius only.

The real part of RI becomes too low when depolarization is added. This effect was discussed in our previous publications (Veselovskii et al., 2010; Muller et al., 2013). In this study, adding depolarization to 29 March measurements decreased mR in dust layer from 1.50 to 1.45, which is too low. The account for spectral dependence of m(I), in opposite, increases mR up to 1.54. Still difference between derived mR is less than estimated uncertainty of retrieval ±0.05.

Corresponding comment is added in the revised manuscript.

*32.        Page 17, line 3. One of the conclusions is that for small enough depolarization the 3-backscatter + 2-extinction + 1-depolarization inversion permits the spherical/nonspherical fraction to be estimated. This isn't part of the analysis of the paper and there is no real support for it here; it might be better to delete it. If it is kept, then besides supporting it with further analysis, it would also be good to clarify whether this gives more complete or more accurate information than the spherical/non-spherical separation that has been practiced for lidar measurements for over a decade (Sugimoto and Lee, 2006; Tesche et al. 2009)*

We agree with the reviewer, this part has been removed.

*33. Figure 1, caption. What quantity is the "lidar signal"?*

It is arbitrary units. Added to figure capture.

*34. Figure 3, caption and annotation. It would be useful to explain the correspondence between the four trajectories and the regions A,B,C,D from Figure 1.*

Added to the fig.3 capture: "First two back trajectories correspond layer A from fig.1, while last two back trajectories correspond layer B from the same figure."

TECHNICAL COMMENTS

Page 16, line 20. "Somehow" = "somewhat"

Done

Figure 4, the labels on the color axis are too small.

Increased

REFERENCES

Tesche, M., A. Ansmann, D. Müller, D. Althausen, R. Engelmann, V. Freudenthaler, and S. Groß (2009), Vertically resolved separation of dust and smoke over Cape Verde using multiwavelength Raman and polarization lidars during Saharan Mineral Dust Experiment 2008, J. Geophys. Res., 114(D13), D13202.

It was in the list

---

## Author Comment (AC2) · 4 May 2016

*Major Comments:*
*1. The title: You should mention the "inversion" in the title, so it*
*becomes more specific. Take care with the title, especially when you are planning a second part*
*of the "SHADOW" campaign.*

Changed for:

Retrieval of optical and physical properties of African dust from multi-wavelength Raman lidar measurements during the "SHADOW" campaign in Senegal

*2. Chapter 3 "Troposphere Stratification and Dynamics": It is not linked with rest of the manuscript and it is not mentioned in the conclusion. Maybe you should prepare a separate publication dealing with wind information; there you may use the information of the micropulse lidar, too. It was not used for this publication, although it provides depolarization information down to 300m. Or you should present one of your three case studies in this detailed manner. But in total it is still a very nice measurement case to introduce the possibilities of your instruments.*

We understand the reviewer's point of view. Nevertheless,  we prefer to keep this section as it is for illustrating the capability of the joint use of both instruments. A separate publication on synergy of wind and Raman lidars is ongoing using extended data sets.

*3. Your LILAS lidar: Is there a reference describing the lidar system in more detail? A well characterized lidar system is crucial for the data quality. Why you use 47 deg I assume that your entire lidar system is inclined by 47_. Have you measured the transmission ratios (transmittance of cross and parallel polarized light) of your detection unit as they may affect the total signal in presence of heavy dust plumes with high depolarization ratios?*

No, we didn't publish anything about LILAS yet. A description of the system is under review in a more "technical" journal (AMT). Since the paper is not accepted yet (and could be rejected), we decided to provide a limited amount of informations.

Measurements were performed from inside the IRD building through a window. 47 deg to horizon was the maximal angle possible

Reviewer is right, it is important for dust plumes., We compared transmission at the two polarizations. We have opportunity to use polarization cubes for all three elastic channels in the receiver and after calibration to combine co- and cross-polarized components.

We have added some information about LILAS in the text.

*4. Comparing your results to the AERONET retrieval (p16l25) is not the final proof as AERONET is using a particle shape model, too. So AERONET is affected by nonspherical particles with high depolarization ratio, too. It would be better to compare your values to in situ measurements. So I suppose using the depolarization information from a lidar system would improve the inversion.*

 AERONET is not the final proof  for sure but it fits the whole phase function while difference between spheres and non spheres is significant for large scattering angles. But we agree with reviewer, that problem of validation is the most critical for lidar inversion.

Minor Comments:

*p4l7 "demands the use of assumptions"*

Corrected

*p5l21 add the year to the date (2015)*

Added

*p7l19 add the year to the date (2015)*

Added

*p8l20-21 "to at" something is missing*

Corrected

*p12 chap "10 April" You have not considered the lidar ratio. Please comment on it while classifying the aerosol layers.*

The sentence is added

*p13l15 "in a first guess" (missing space)*

Corrected

*p13l18 15_m as a maximum particle radius seems small so close to the desert.*

Yes, it would be good to consider larger radii, but maximal wavelength available for us is 1.064 µm and for intervals above 15 µm the retrieval becomes unstable since the sensitivity of the measurements to large particles is very limited.

*fig 1: add the year to the date (2015); add a description to the color bar;*

Added

*fig 2: mention "horizontal" wind to not get confused with vertical winds*

Added

*fig 4: Wouldn't you expect higher depolarization values for the more continental air (CT)? Lidar ratios for marine particles in Layer A are very high. Could you please comment on this?*

Similar comment we had from Reviewer 1. The air masses which call CMT still contain a lot of dust, so it is not pure maritime aerosol and depolarization is quite high through the whole layer A. We have added the comment to the manuscript.

*Fig 6: "Typically EAE varies in 0-0.3 range, but during dust episodes the values of EAE became negative, decreasing to _-0.15." It is not seen in the insert of fig 6, as nearly all cases of EAE are in the Angström box "0". You may choose a different scaling for the insert diagram.*

The scale of figure is chosen to show that EAE is mainly inside [-0.2, 0.2] interval. Uncertainty of EAE calculation is about 0.2, so we think there is no reason to plot statistics with higher resolution. Nevertheless, we slightly modified the text, removing "Typically EAE varies in 0-0.3 range…" because negative EAE is also typical for dust.

*Fig 7: To see the variability in the particle depolarization a separate diagram for the depolarization values would be nice. There is enough space to put error bars to every*

*depolarization value without confusing the reader, so it would be a good idea to add the error bars.*

Variation of depolarization is in 20%-35% range, so for such small interval, a separate figure won't bring additional informations.

Although there is enough space to add uncertainties for depolarization, we would need to add also uncertainties for lidar ratios also , which would result in a confusing plot

*Fig 9a: beta 1064 should be shown in red as in the other plots.*

Corrected

*Fig 14: Are you sure with the peak of beta532 at 3 km? See description: there are no open symbols (this belongs to fig 15, I suppose).*

To be exact the peak is at 3039 m, so added "approximately".

Yes, it was from fig.15. Removed.

---

## Author Comment (AC3) · 4 May 2016

*1.	The authors conclude that the negative values of the BAEs measured for dust are due to the enhanced absorption in the UV. This is not supported by independent measurements. It is also well-known that the spheroid model assumes a spectrally independent phase function at 180 degrees. A possible spectral dependence on the 180 phase function could also be the source of negative BAEs and this limitation of the spheroid model should be mentioned in the manuscript and in the conclusion section.*

It is true, we had no available independent measurements of dust refractive index. However such measurements we done during SAMUM campaigns in West Africa. So spectral dependence of imaginary part in our measurements looks very probable.

Phase function in spheroid model depends on size parameter, in this way it is spectrally dependent.

*2. The paper gives the impression that the particle depolarization ratio does not provide significant information on the inversion. However, there is much discussion in the literature (see for example the work of Gasteiger) that the spheroid model cannot reproduce the lidar measurements of the linear particle depolarization ratio. Thus, how we expect that an inversion code based on the spheroid model would show that there is an added value on the microphysical retrievals by adding depolarization information? I think that the conclusions should be rephrased, such as to make clear that this could be a limitation of the spheroid model as well.*

This question was posed also by Reviewer 1. Yes, spheroid model has issues in reproducing depolarization measurements, though it is not easy to quantify these without laboratory measurements in chamber. We added several comments in the text, in particular that  results presented should stimulate development of the dust model with improved capability to mimic dust depolarization properties.

---

## Author Response (AR2)

Major Issues

*- I agree with Referee #2 that Section 3 is not necessary to convey the message of the paper. It should be omitted; particularly as no optical profiles are presented for this measurement period. Just illustrating the capability of joint measurements does not warrant publication.*

As requested, we changed the structure of manuscript. Section 3 is removed. Instead, we added a sub-section to
Section 2 which describes the meteorological conditions during the measurements and inserted there fig.1-4. We have added also Fig.5, showing the correlation of profiles of extinction and water vapor.
To the plots describing results on 29 March and 10 April we have added profiles of water vapor (for 13 March vapor measurements are not available).

*- The discussion on page 15, lines 11-21 is not clear. Why are results of 3b+2a more trustworthy than those of 3b+2a+1d? This is an important issue that should be discussed in more detail, even if you already addressed it elsewhere. In the latter case a review of key findings will do.*
We made revision of this paragraph, now it should be more clear.

- Please *improve the quality of your figures.*
*\* I suggest to use two sub-figures instead of insets in Figures 6, 7, 12, and 16.*
We have done it for fig.6, 7, 16 but for fig.12 we believe the sub-figure should be kept as an insert for clarity. Please notice that we have added one new figure (fig.5) and removed one (fig.15), so the numeration of figures is shifted.
*\* Add error bars to all points in Figures 6 and 7.*
Done
*\* Use another color for the blue line in Figures 8, 10, and 13 since blue is unrecognizable over water.*
Done
*Also add symbols for the time intervals to the trajectories in the upper part of the figures.*
This information is in the bottom of the plots and included in the caption to the figures.
*\* Use the same color for lines and error bars in Figures 9, 11, and 14.*

Done

Minor Issues

*- Make sure that your answers to points 14, 18, 19 and 27 of Referee #1 are also mentioned in the manuscript.*

Added

*- Don't use R as symbol for the lidar ratio as it might be confused with the backscatter ratio (and you also use LR in the text).*

Corrected for LR

*- Use the terms backsatter coefficient and extinction coefficient instead of just backscatter and extinction (also in Figures).*

Corrected

*Always provide the wavelengths pair for the Angstroem exponent (also in Figures).*

There is not enough space in figures to show wavelengths, so we do it in captions to the figures.

*- When discussion the complex refractive index measured during SAMUM you should refer to T. Mueller et al. (2009): Spectral absorption coefficients and imaginary parts of refractive indices of Saharan dust during SAMUM-1, Tellus B, 10.1111/j.1600-0889.2008.00399.x*

Reference is added

*- mention the availability of AERONET measurements already in Section 2*

Done

*- Figure 15 and its discussion are unnecessary and could be omitted.*

Removed